# Allosteric Determinants of the SARS-CoV-2 Spike Protein Binding with Nanobodies: Examining Mechanisms of Mutational Escape and Sensitivity of the Omicron Variant

**DOI:** 10.3390/ijms23042172

**Published:** 2022-02-16

**Authors:** Gennady Verkhivker

**Affiliations:** 1Keck Center for Science and Engineering, Graduate Program in Computational and Data Sciences, Schmid College of Science and Technology, Chapman University, Orange, CA 92866, USA; verkhivk@chapman.edu; Tel.: +1-714-516-4586; 2Department of Biomedical and Pharmaceutical Sciences, Chapman University School of Pharmacy, Irvine, CA 92618, USA

**Keywords:** SARS-CoV-2 spike protein, ACE2 host receptor, nanobodies, molecular dynamics, mutational sensitivity, binding free energy, allosteric interactions, signal transmission

## Abstract

Structural and biochemical studies have recently revealed a range of rationally engineered nanobodies with efficient neutralizing capacity against the SARS-CoV-2 virus and resilience against mutational escape. In this study, we performed a comprehensive computational analysis of the SARS-CoV-2 spike trimer complexes with single nanobodies Nb6, VHH E, and complex with VHH E/VHH V nanobody combination. We combined coarse-grained and all-atom molecular simulations and collective dynamics analysis with binding free energy scanning, perturbation-response scanning, and network centrality analysis to examine mechanisms of nanobody-induced allosteric modulation and cooperativity in the SARS-CoV-2 spike trimer complexes with these nanobodies. By quantifying energetic and allosteric determinants of the SARS-CoV-2 spike protein binding with nanobodies, we also examined nanobody-induced modulation of escaping mutations and the effect of the Omicron variant on nanobody binding. The mutational scanning analysis supported the notion that E484A mutation can have a significant detrimental effect on nanobody binding and result in Omicron-induced escape from nanobody neutralization. Our findings showed that SARS-CoV-2 spike protein might exploit the plasticity of specific allosteric hotspots to generate escape mutants that alter response to binding without compromising activity. The network analysis supported these findings showing that VHH E/VHH V nanobody binding can induce long-range couplings between the cryptic binding epitope and ACE2-binding site through a broader ensemble of communication paths that is less dependent on specific mediating centers and therefore may be less sensitive to mutational perturbations of functional residues. The results suggest that binding affinity and long-range communications of the SARS-CoV-2 complexes with nanobodies can be determined by structurally stable regulatory centers and conformationally adaptable hotspots that are allosterically coupled and collectively control resilience to mutational escape.

## 1. Introduction

SARS-CoV-2 infection is transmitted when the viral spike (S) glycoprotein binds to the host cell receptor ACE2, leading to the entry of S protein into host cells and membrane fusion [1,2]. The full-length SARS-CoV-2 S protein consists of amino (N)-terminal S1 subunit and carboxyl (C)-terminal S2 subunit where S1 is involved in the interactions with the host receptor and includes an N-terminal domain (NTD), the receptor-binding domain (RBD), and two structurally conserved subdomains (SD1 and SD2). Structural and biochemical studies established that the mechanism of virus infection may involve conformational transitions between distinct functional forms of the SARS-CoV-2 S protein in which the RBDs continuously switch between “down” and “up” positions [3,4,5,6,7,8,9,10,11,12]. The SARS-CoV-2 antibodies are divided into several main classes, of which class 1 and class 2 antibodies target epitopes that overlap with the ACE2 binding site [13,14,15]. The body of structural and biochemical studies of the SARS-CoV-2 S complexes with different classes of potent antibodies targeting distinct binding epitopes of the S-RBD as well as various antibody cocktails and combinations have revealed multiple conformation-dependent epitopes, highlighting the link between conformational plasticity and adaptability of S proteins and capacity for eliciting specific binding and broad neutralization responses [16,17,18,19,20,21,22,23,24,25,26,27,28,29,30,31,32]. These studies have examined SARS-CoV-2 S binding with antibodies showing that combinations of antibodies can provide efficient cross-neutralization effects through synergistic targeting of conserved and variable SARS-CoV-2 RBD epitope. Structural studies confirmed that the SARS-CoV-2 S protein could feature distinct antigenic sites, and some specific antibodies may allosterically inhibit the ACE2 receptor binding without directly interfering with ACE2 recognition [29]. Optimally designed antibody cocktails simultaneously targeting different binding epitopes on the SARS-CoV-2 RBD also demonstrated improved resilience against mutational escape [33,34,35].

Nanobodies or single-domain antibodies provide important advantages over traditional antibodies, including their smaller size and robust biochemical properties such as high thermal stability, high solubility, and ability to be bioengineered into novel multivalent, multi-specific, and high-affinity molecules, making them a class of emerging powerful therapies against SARS-CoV-2 [36,37,38,39,40,41]. Recent research efforts in the design, engineering, and structure-functional characterization of nanobodies and their binding with SARS-CoV-2 S proteins reflect a growing realization that nanobody combinations could deliver a powerful array of neutralizing and escape mutation resistant molecular assemblies capable of rationally exploiting distinct binding epitopes and the intrinsic plasticity of the SARS-CoV-2 S protein. Structural aspects and classification of the nanobodies binding with the SARS-CoV-2 S were recently discussed in a review [42], highlighting several classes of high-affinity nanobodies 

An ultra-potent synthetic nanobody, Nb6, neutralizes SARS-CoV-2 by stabilizing the fully inactive down S conformation preventing binding with ACE2 receptor [43]. A high-affinity trivalent nanobody, mNb6-tri, can simultaneously bind to all three RBDs and inhibit the interactions with the host receptor by occupying the binding site and locking the S protein in the inactive state [43]. The size-exclusion chromatography and mass spectrometry revealed high-affinity RBD-targeting nanobodies that efficiently neutralize SARS-CoV-2 by using several distinct and non-overlapping epitopes [44]. The revealed dominant epitope targeted by Nb20 and Nb21 nanobodies overlaps with the ACE2 binding site, showing that these nanobodies could competitively inhibit ACE2 binding and exploit structural mimicry to facilitate conformational changes that prematurely convert spike into a post-fusion state suppressing viral fusion [44]. Potent neutralizing nanobodies that resist circulating variants of SARS-CoV-2 by targeting novel epitopes were recently discovered [45]. The reported cryo-EM structures for different classes of nanobodies suggested mechanisms of high-affinity and broadly neutralizing activity by exploiting epitopes that are shared with antibodies as well as novel epitopes that are unique to the nanobodies [45]. The high-affinity nanobodies against SARS-CoV-2 S protein refractory to common escape mutants and exhibiting synergistic neutralizing activity are characterized by proximal but non-overlapping epitopes showing that multimeric nanobody combinations can improve potency while minimizing susceptibility to escape mutations [46]. These studies identified a group of common resistant mutations in the dynamic RBM region (F490S, E484K, Q493K/R, F490L, F486S, F486L, and Y508H) that evade many individual nanobodies. Structural versatility of nanobody combinations that can effectively insulate the S-RBD accessible regions suggested a mechanism of resistance to mutational escape in which combining two nanobodies can markedly reduce the number of allowed substitutions to confer resistance and thereby elevate the genetic barrier for escape [46,47]. Using human VH-phage library and protein engineering, several unique VH binders were discovered that recognized two separate epitopes within the ACE2 binding interface with nanomolar affinity [47]. Multivalent and bi-paratopic VH constructs showed markedly increased affinity and neutralization potency to the SARS-CoV-2 virus when compared to the standalone VH domain [47]. Using saturation mutagenesis of the RBD exposed residues combined with fluorescence-activated cell sorting for mutant screening, escape mutants were identified for five nanobodies and were mostly mapped to the periphery of the ACE2 binding site, with K417, D420, Y421, F486, and Q493 emerging as notable hotspots [48]. A wide range of rationally engineered nanobodies with efficient neutralizing capacity and resilience against mutational escape was recently unveiled that included the llama-derived nanobody VHH E bound to the ACE2- binding epitope and three alpaca-derived nanobodies, VHHs U, V, and W, that bind to a different cryptic RBD epitope [49]. Using X-ray crystallography and surface plasmon resonance-based binding competition, this study showed that combinations of nanobodies targeting distinct epitopes could suppress the escape mutants resistant to individual nanobodies, while the bi-paratopic VHH EV and VE nanobodies with two antigen-binding sites appeared to be even more effective than pairs VHH E+U, E+V, and E+W in preventing mutual escape [40,41,49]. Using single-domain antibody library and PCR-based maturation, two closely related and highly potent nanobodies, H11-D4 and H11-H4, were reported that recognize the same epitope immediately adjacent to and partly overlapping with the ACE2 binding region [50]. The crystal structures of these nanobodies bound to the S-RBD revealed binding to the same epitope, which partly overlaps with the ACE2 binding surface, explaining competitive inhibition of ACE2 interactions. These studies demonstrated that nanobodies might have potential clinical applications due to the increased neutralizing activity and robust protection against escape mutations of SARS-CoV-2. 

The high-affinity nanobody cocktails of two noncompeting nanobodies can neutralize both wild-type SARS-CoV-2 and the variants [51]. Neutralization of SARS-CoV-2 by low-picomolar and mutation-tolerant VHH nanobodies that bind synergistically to the opposite sides of the RBD produced a binding avidity effect unaffected by immune-escape mutants K417N/T, E484K, N501Y, and L452R [52]. The nanobody cocktails from camelid mice and llamas that neutralize SARS-CoV-2 variants showed a remarkable ability of multivalent nanobodies to combat escaping mutations through synchronized avidity between binding epitopes. In particular, picomolar nanobodiesNb12 and Nb30 revealed binding to a conserved RBD epitope outside of the ACE2-binding motif, which is not accessible to human antibodies allowing for combat escape mutations at E484 and N501 positions [53]. These studies suggested that nanobody mixtures and rationally engineered bi-paratopic nanobody constructs could offer a promising alternative to conventional monoclonal antibodies and may be advantageous for controlling a broad range of infectious variants while also suppressing the emergence of virus escape mutations. Furthermore, bi-paratopic nanobodies showed significant advantages compared to monoclonal antibodies, single nanobodies, and nanobody cocktails by effectively leveraging binding avidity and allosteric cooperativity mechanisms in combating escape mutations. The recent biophysical studies indicated that avidity-driven mechanisms might underlie functional effects of nanobody combinations and multivalent nanobody constructs to prevent viral escape making it possible to rationally engineer desirable levels of binding specificity and generation of ultra-potent molecules for targeting SARS-CoV-2 S proteins. Avidity-inspired nanobody therapeutics can leverage the emerging evidence of how binding affinity, avidity, and cooperativity are balanced in a complex thermodynamic mechanism of synchronous binding of multivalent nanobody constructs [38]. 

The emergence of variants of concern (VOCs) with the enhanced transmissibility and infectivity profile including the D614G variant [54,55,56,57], B.1.1.7 (alpha) [58,59,60,61], B.1.351 (beta) [62,63], B.1.1.28/P.1 (gamma) [64], and B.1.1.427/B.1.429 (epsilon) variants [65,66] have attracted enormous attention in the scientific community and a considerable variety of the proposed mechanisms explaining functional observations from structural and biochemical perspectives. The detection of common mutational changes such as D614G, E484K, N501Y, and K417N that are shared among major circulating variants B.1.1.7, B.1.351, and B.1.1.28/P.1 indicated that these positions could be particularly critical for modulation of the SARS-CoV-2 S protein responses. Biophysical studies of the SARS-CoV-2 S trimers for these variants revealed structural and functional effects of mutations that can modulate dynamics and stability of the closed and open forms, increase binding to the human receptor ACE2, and confer immunity escape from vaccines and different classes of monoclonal antibodies and nanobodies [67,68,69,70,71]. 

The recent VOC, omicron (B.1.1.529), displaying a large number of mutations in the S-RBD regions, has further intensified the scientific and public interest and concerns about the role and mechanisms underlying the emergence of variants [72,73,74,75,76]. The latest structural and biophysical tour-de-force investigation convincingly demonstrated that Omicron-B.1.1.529 mutational diversity could induce a widespread escape from neutralizing antibody responses [75]. According to this study, mutations S477N, Q498R, and N501Y increase ACE2 affinity by 37-fold, serving to anchor the RBD to ACE2, while allowing the RBD region freedom to develop further mutations, including those that reduce ACE2 affinity in order to evade the neutralizing antibody response [75]. Strikingly, K417N, T478K, G496S, Y505H, and the triple S371L, S373P, S375F can reduce affinity to ACE2 while driving immune evasion and providing a final net affinity for ACE2 similar to the original virus. Structural studies examined several VOCs and demonstrated that Omicron variant RBD binds to human ACE2 with comparable affinity to that of the original virus [76]. The crystal and cryo-EM structures of Omicron RBD complexed with human ACE2 identified the role of key residues for receptor recognition showing that mutations E484A, Q493R, and Q493R are responsible for immune escape from monoclonal antibodies.

Biophysical studies provided an enormous insight into the mechanisms underlying differential binding of the S protein variants to the host receptor ACE2 and antibodies. A series of illuminating biophysical investigations analyzed the biophysical properties of the SARS-CoV-2 S-glycoprotein binding to ACE2 on model surfaces and on living cells using force–distance (FD) curve-based atomic force microscopy (FD-curve-based AFM) [77,78]. By using atomic force microscopy and computer simulations, the kinetic and thermodynamic parameters of binding between the ACE2 receptors on the model surface and S-RBD variants (Alpha, Beta, Gamma, and Kappa) were investigated [78]. By providing unprecedented atomistic-level details and significant insight into molecular binding mechanisms of the SARS-CoV-2 variants, this study observed that the N501Y and E484Q mutations are particularly important for the greater stability, while the N501Y mutation is unlikely to significantly affect antibody neutralization [78]. By probing the interactions using AFM force spectroscopy, it was shown that the RBD mutations in different variants typically result in the higher stability and affinity of the complex with ACE2, which can mediate the increased transmissibility [78]. Moreover, integration of biophysical experiments and molecular simulations support the idea of a stabilized interface through multiple weaker molecular interactions that cooperatively stabilize the interface between the RBD and the ACE2 receptor.

Computer simulations and protein modeling also played an important role in shaping our understanding of the dynamics and function of SARS-CoV-2 glycoproteins [79,80,81,82]. All-atom molecular dynamics (MD) simulations of the full-length SARS-CoV-2 S glycoprotein embedded in the viral membrane, with a complete glycosylation profile, were first reported by Amaro and colleagues, providing an unprecedented level of details and significant structural insights about functional S conformations [81,82]. A simplified model of the SARS-CoV-2 virion integrated data from cryo-EM, x-ray crystallography, and computational predictions to build molecular models of structural SARS-CoV-2 proteins assemble a complete virion model [83]. Multi-microsecond MD simulations of a 4.1 million atom system containing a patch of viral membrane with four full-length, fully glycosylated and palmitoylated S proteins allowed for a complete mapping of generic antibody binding signatures and characterization of the antibody and vaccine epitopes [84]. MD simulations and free energy landscape mapping studies of the SARS-CoV-2 S proteins and mutants detailed conformational changes and diversity of ensembles, further supporting the notion of enhanced functional and structural plasticity of S proteins [85,86,87,88,89,90,91]. Using data analysis and protein structure network modeling of MD simulations, residues that exhibit long-distance coupling with the RBD opening, including sites harboring functional mutations D614G and A570D, which points to the important role of the D614G variant in modulating allosteric communications in the S protein [87]. The free energy landscapes of the S protein derived from MD simulations together with nudged elastic pathway optimization mapping of the RBD opening revealed a specific transient allosteric pocket at the hinge region that is located near the D614 position influences RBD dynamics [88]. 

Computational and biophysical kinetics studies of the SARS-CoV-2 S trimer interactions with ACE2 using the recent crystal structures also provided important insights into the key determinants of the binding affinity and selectivity [92,93,94,95]. Our recent studies combined simplified and atomistic MD simulations with coevolutionary analysis and network modeling to present evidence that the SARS-CoV-2 spike protein function as an allosterically regulated machine that exploits the plasticity of allosteric hotspots to fine-tune response to antibody binding [96,97,98,99,100,101,102,103,104,105]. These studies showed that examining the allosteric behavior of the SARS-CoV-2 spike proteins may be useful to uncover functional mechanisms and rationalize the growing body of diverse experimental data. 

Using MD simulations and protein stability analysis, we recently examined binding of the SARS-CoV-2 RBD with single nanobodies Nb6 and Nb20, VHH E, a pair combination VHH E+U, a bi-paratopic nanobody VHH VE, and a combination of CC12.3 antibody and VHH V/W nanobodies [105]. This study characterized the binding energy hotspots in the SARS-CoV-2 protein and complexes with nanobodies providing a quantitative analysis of the effects of circulating variants and escaping mutations on binding that is consistent with a broad range of biochemical experiments. The results suggested that mutational escape may be controlled through structurally adaptable binding hotspots in the receptor-accessible binding epitope that are dynamically coupled to the stability centers in the distant binding epitope targeted by VHH U/V/W nanobodies [105]. Using computer-based design of protein–protein interactions, a number of nanobodies were engineered in silico and selected based on the free energy landscape of protein docking verified by the recently reported cocrystal structures [106]. Another computational study examined binding mechanisms of neutralizing nanobodies targeting SARS-CoV-2 S proteins [107]. All-atom MD simulations totaling 27.6 μs in length using the recently solved structures of the RBD of SARS-CoV-2 S protein in complex with nanobodies H11-H4, H11-D4, and Ty1 revealed interactions between S-RBD and the nanobodies and estimated that the binding strength of the nanobodies to RBD is similar to that of ACE2 [107].

In the present work, we expanded the analysis of the SARS-CoV-2 S protein binding with nanobodies by performing a large number of high resolution coarse-grained (CG) simulations followed by full atomistic reconstruction for the complete S protein trimer complexes with multivalent nanobodies Nb6, VHH E, and VHH E/VHH V nanobodies. In addition, we also performed all-atom MD simulations and provided a detailed comparative analysis of conformational dynamics profiles for the S trimer complexes with the examined panel of nanobodies. Atomistic dynamics and analysis of collective motions are combined with a battery of computational tools to examine energetics and allosteric interactions, including binding free energy scanning, perturbation-response scanning, and network modeling. Through the synergistic application of these simulation methods, we examine the atomic-level mechanisms of binding-induced allosteric modulation in the SARS-CoV-2 S trimer complexes with nanobodies. By quantifying energetic and allosteric determinants of the SARS-CoV-2 S binding with nanobodies, we also analyze the effects of escaping mutations and the effect of the Omicron variant mutations on nanobody binding. The results suggest that binding affinity and allosteric signatures of the SARS-CoV-2 complexes can be determined by a dynamic cross-talk between structurally stable regulatory centers and conformationally adaptable allosteric hotspots that collectively control resilience to mutational escape. 

## 2. Results and Discussion

### 2.1. Conformational Dynamics and Collective Motions of the SARS-CoV-2 S Trimer Complexes: Nanobody-Induced Modulation of Flexibility and Escape Mutation Sites as Regulatory Hinges

We performed multiple CG simulations of the SARS-CoV-2 S trimer protein complexes with a panel of nanobodies (Figure 1) followed by all-atom reconstruction of trajectories to examine how structural plasticity of the RBD regions can be modulated by binding and determine specific dynamic signatures induced by different classes of nanobodies targeting distinct binding epitopes. All-atom MD simulations with the explicit inclusion of the glycosylation shield could provide a rigorous assessment of the conformational landscape of the SARS-CoV-2 S proteins; such direct simulations remain technically challenging due to the size of a complete SARS-CoV-2 S system embedded onto the membrane. We combined CG simulations with atomistic reconstruction and additional optimization by adding the glycosylated microenvironment. CG-CABS trajectories were subjected to atomistic reconstruction and refinement. In addition, and for a direct comparative analysis, we also performed all-atom MD simulations of the S trimer complexes with nanobodies. Using a comparison of CG-CABS and MD simulations, we verified the reliability of the proposed simulation model and examined how SARS-CoV-2 spike protein can exploit the plasticity of the RBD regions to modulate specific dynamic responses to nanobody binding. The conformational dynamics profiles for CG-CABS simulations describe the mean residue-based thermal fluctuations averaged over 100 independent CG simulations (Figure 2). 

A comparative analysis of the conformational flexibility profiles for the S trimer complexes with Nb6, VHH E, and VHH E/VHH V nanobodies revealed stabilization of the interacting regions that was particularly strong in the complex with the VHH E/VHH V nanobody pair (Figure 2A). The RBD core α-helical segments (residues 349–353, 405–410, and 416–423) showed small thermal fluctuations in all complexes. The stability of the central β strands (residues 354–363, 389–405, and 423–436) was especially pronounced in the S trimer complex with Nb6 nanobody (Figure 2A). Both CG-CABS and all-atom MD simulation models reproduced the overall stability of the conserved S-RBD core formed by antiparallel β strands (β1 to β4 and β7) (residues 354–358, 376–380, 394–403, 431–438, 507–516) (Figure 2 and Figure 3). Atomistic MD simulations also showed only moderate fluctuations of β5 and β6) (residues 451–454 and 492–495) that connect the mobile RBM region to the central core (Figure 2). The results showed that Nb6 binding to the closed conformation of the S trimer could induce a more significant stabilization of the S-RBD and RBM residues (Figure 2A). 

Interestingly, all-atom MD simulations of the SARS-CoV-2 S trimer bound to Nb6 revealed a more significant mobility of the RBD regions as compared to the conformational profile obtained in the CG-CABS simulations (Figure 2B). A greater level of flexibility was seen in CG-CABS and atomistic MD simulations for the S-RBD regions in the S trimer complexes with VHH E (Figure 2C) and VHH E/VHH V nanobodies (Figure 2D). Hence, the conformational plasticity of the RBD-up conformations can still be maintained in the complexes with nanobodies. In comparison with all-atom MD trajectories, the CG-CABS model produced higher average residue oscillations, which is consistent with the previous validation studies of the CABS model [109]. Consistently, both CG-CABS and all-atom MD simulations highlighted the greater stability of the highly conserved S2 subunit (residues 686–1162) as compared to a more adaptable S1 subunit that includes NTD (residues 14–306), RBD (residues 331–528), CTD1 (residues 528–591), and CTD2 (residues 592–685) (Figure 2). In particular, all-atom MD simulations of the S trimer complex with VHH E nanobody showed a more significant difference in the stabilization of the S1 and S2 domains by displaying very small fluctuations in the S2 regions and larger fluctuations of the S1 regions. (Figure 2C). Although the VHH E epitope is very similar to that of other nanobodies in this class, such as Nb6, VHH E binds in a specific orientation in which an extended β-hairpin conformation protrudes into the RBD binding site (Figure 2B,C). Conformational dynamics profiles reaffirmed stability of the α-helical segments in the RBD that are located near the cryptic binding epitope (residues 369–384) targeted by the VHH V nanobody. Importantly, binding of the VHH V nanobody to the cryptic epitope restricted mobility of the S2 subunit residues (Figure 2A). Based on these observations, we argue that these residues could provide a stable anchoring platform at the cryptic epitope for the VHH V nanobody, while allowing for optimization of binding interactions with the more dynamic RBD binding epitope (Figure 2). 

To highlight similarities and differences in the mobility profiles derived from CG-CABS and all-atom MD simulations, we performed a simple statistical analysis and computed averages and standard deviations of the RMSF values. In addition, to compare CG-CABS and all-atom MD trajectories and establish a correspondence between the dynamics profiles produced through atomistic reconstruction of CG-CABS trajectories and all-atom MD simulations, we computed the average Spearman’s correlation coefficient (*r_s_*) between the respective RMSF profiles. Given the differences between these simulation models, the correlation analysis confirmed a similar pattern of protein flexibility, yielding statistically significant correlation *r_s_* = 0.68 for the S trimer complexes with Nb6, *r_s_* = 0.723 for the S trimer complexes with VHH E, and only slightly lower *r_s_* = 0.624 for the complex with VHH E/VHH V nanobody. These results are similar to the outcome of the large-scale validation study that yielded the average Spearman’s correlation coefficient of *r_s_* ~ 0.7 between the RMSFs of the CG-CABS and atomistic simulations for the diverse protein set [109]. Interestingly, this study also showed that correlations among MD trajectories obtained from different all-atom force fields could vary in a similar range (0.75–0.82) [109]. The observed similarities of the conformational dynamics profiles suggested that CG-CABS simulations could provide a fairly accurate and affordable simulation approach for quantifying flexibility of the SARS-CoV-2 S complexes with the panel of nanobodies. In general, our results supported the previous studies [109], indicating that atomistic reconstruction of CG-CABS trajectories could produce adequate protein flexibility profiles that are consistent with all-atom simulations and, due to a much lower cost, allow for multiple independent runs and accumulation of statistically significant averages.

Structural maps of the conformational dynamics profiles for the S-RBD complexes with Nb6 (Figure 3A), VHH E (Figure 3B), and VHH VE (Figure 3C) illustrated an appreciable mobility of the NTD and RBD residues in the 3-up complexes with VHH E and VHH E/VHH V nanobodies. The closed conformation of the S trimer complex with Nb6 is more rigid (Figure 3A), but an appreciable level of mobility could be seen in the S1 subunit NTD and RBD regions. The results showed that the open state of the S trimer bound to VHH E nanobody (Figure 3B) with all RBDs in the up position are generally more flexible in the S1 regions, while structural rigidity of the S2 regions becomes even more pronounced for these states. Accordingly, collective movements of the S1 regions anchored by the rigid S2 core could become more pronounced in the more dynamic open states, allowing for large rigid body movements of the NTD and RBD regions. 

This dynamics pattern is consistent with the notion that single nanobody binding to the ACE2 binding site can only partly restrict the intrinsic mobility of the RBD regions, allowing for conformational adaptability and potential escape from neutralization. Interestingly, the conformational dynamics map of the open S trimer complex with VHH E/VHH V nanobodies showed a more significant rigidification of the entire S protein, including both S1 and S2 subunits (Figure 3C). Although the RBDs are in the up position, nanobody binding at two distinct epitopes can impose more severe restrictions on the RBD movements and arguably allow for more effective inhibition of the RBD-ACE2 interactions.

We characterized collective motions for the SARS-CoV-2 S-RBD complexes averaged over low-frequency modes using principal component analysis (PCA) of the trajectories (Figure 4). The local minima along these profiles are typically aligned with the hinge centers, while the maxima correspond to the moving regions undergoing concerted movements. The low-frequency ‘soft modes’ are often functionally important as mutations or binding can exploit and modulate protein movements along the pre-existing slow modes to induce allosteric transformations. The overall shape of the essential profiles in the SARS-CoV-2 S trimer complex with Nb6 showed suppressed movements of RBDs that are in the down position (Figure 4A,B). On the other hand, the profile displayed larger functional displacements of the NTD regions. The immobilized hinge positions of the S trimer corresponded to positions F318, L387, F429. The slow mode profile of the S trimer complex with Nb6 showed the reduced RBD mobility, but the tip of the RBM loop (residues 473–483) remained mobile in functional dynamics. The sites of typical nanobody-escaping mutations (G447, Y449, L452, F490, Q493, Y508) correspond to the low mobility RBD regions in slow modes of the S trimer (Figure 4A,B). Although the RBD region harboring E484/F486 positions undergoes some functional motions in the slow modes, these movements are relatively moderate as compared to the NTD fluctuations that dominate collective dynamics. Nb6 binding could be severely compromised by the E484K mutation, while other sites of nanobody-escaping mutations are likely to be suppressed by the nanobody [43]. This may be partly explained based on the functional dynamics profiles in which most of these positions are immobilized by Nb6 binding, whereby the absence of functional motions could restrict the mutational escape potential. The fact that only the tip of the RBM region and E484/F486 remain more prone to changes could allow for E484K mutation to escape Nb6 binding and adopt a conformation evading efficient nanobody interactions. The slow mode profile of the S trimer complex with VHH E nanobodies in which all RBDs are in the up position showed a clearly different pattern (Figure 4C,D). In this case, the RBDs correspond to moving regions. The rigid hinge centers are located at conserved F318 and V534, F592 residues. Several local hinge positions are aligned with I358, A363, Y365, L387 in the RBD core due to constraints imposed by RBD interactions with NTD of the adjacent protomer. The local maxima of the slow mode profile corresponded to V350, V369, S371, F377, K378, G447, Y449, L452, and 476–492 cluster (Figure 4C,D). Some of these functionally mobile residues are not involved in the interactions with VHH E nanobody (V350, V369, S371, F377, K378) and allow for conformational rearrangements of these flexible RBD regions. Instructively, nanobody binding can be partly escaped by mutations Y369H, S371P, F377L, and K378Q/N, even though these modifications are not currently circulating.

Hence, the sites of escaping mutations are aligned with the functionally moving RBD regions, which may experience functional displacements and affect the RBD conformation, thereby reducing the efficiency of VHH E binding. The largest peaks in the slow mode profile are aligned with K417, F456, and RBM residues E484/F486 (Figure 4C,D). Movements of these positions may affect the fidelity of nanobody binding, and mutations in these positions, particularly E484K, can escape the nanobody effect owing to the inherent functional plasticity in this region. This may contribute to a certain level of vulnerability shown by nanobodies Nb6 and VHH E targeting the ACE2-binding site to mutations in K417 and E484 residues. Structural maps of the slow mode profiles for the S complex with VHH E (Figure 4D) illustrate the greater mobility of the RBM residues and plasticity of the binding epitope. A similar picture was observed for the collective dynamics analysis of the S complex with VHH VE nanobody (Figure 4E,F). Our analysis indicated that the VHH VE nanobody could modulate conformational dynamics without dramatically altering collective motions but rather fine-tune dynamic changes at the binding site. These findings are consistent with the experimental evidence showing that VHH E and VHH V nanobodies that target two independent epitopes can activate the SARS-CoV-2 fusion machinery [49]. Although VHH VE binding can curtail flexibility of the S1 regions and impose structural constraints in the binding sites, functional RBD motions are still characteristic of the S complexes may contribute to mutational adaptation as sequences containing mutations in both interfaces were detected in the presence of VHHs E and V [49]. The results may explain why flexible RBD sites F486 and F490 are often featured as common sites of escape mutants that dominate the VHH E interface [49].

### 2.2. Mutational Scanning Identifies Structural Stability and Binding Affinity Hotspots in the SARS-CoV-2 Complexes and Explains Patterns of Nanobody-Escaping Mutations

By employing conformational ensembles of the S trimer complexes with nanobodies, we performed mutational scanning and computed binding free energy changes for studied SARS-CoV-2 S complexes with NB6, VHH E, and combination of VHH E/VHH V. In silico mutational scanning was done using the BeAtMuSiC approach [110,111,112]. This approach allows for accurate predictions of the effect of mutations on both the strength of the binding interactions and on the stability of the complex using statistical potentials and neural networks. This approach showed a comparable performance and accuracy as physics-based FoldX potentials [113,114,115,116]. The BeAtMuSiC approach adapted in our study was further enhanced through ensemble-based averaging of binding energy computations. The binding free energy ΔΔG changes were computed by averaging the results of computations over 1000 samples obtained from simulation trajectories. 

We first analyzed the mutational profiles for the S trimer 3-down complex with Nb6 (Figure 5). Mutational sensitivity analysis of the S binding with Nb6 showed results that were generally consistent with our earlier studies when using MD simulations of the S-RBD complex [105]. In the S trimer complex, however, a single Nb6 molecule is positioned at the interface between two adjacent RBDs (Figure 1) [43]. The experimental studies suggested that a single Nb6 can stabilize two adjacent RBDs in the down state and prime the binding site for a second and third Nb6 molecule to stabilize the 3 RBD-down S conformation [43]. Mutational scanning of the S trimer revealed the binding energy hotspots in each protomer that are distributed through two interfaces, each interacting with a different Nb6 molecule (Figure 5). One of the interfaces corresponded to the cryptic binding RBD site where one Nb6 molecule interacts with N343, V367, S371, S373, V374, W436 hotspots (Figure 5). Our previous studies showed that highly conserved sites F374 and W436 are important coevolutionary centers that are often implicated in interactions with neutralizing antibodies [98,99]. The other Nb6 molecule binds to the ACE2-binding site on the RBD where the key binding energy hotspots corresponded to hydrophobic residues Y449, L453, L455, F456, Y489, F490, G496, and Y505 (Figure 5). A number of these positions are also binding affinity hotspots for ACE2, as evident from deep mutagenesis scanning of SARS-CoV-2 interactions with the ACE2 host receptor [117,118,119,120]. The interaction pattern and similarity in the binding energy hotspots with ACE2 supported the notion of structural mimicry that may be efficiently exploited by Nb6 nanobody to competitively inhibit the ACE binding region. 

The mutational sensitivity map also sheds some light on the structure-functional role of sites targeted by common resistant mutations (F490S, E484K, Q493K/R, F490L, F486S, F486L, and Y508H) that evade many individual nanobodies [46]. Indeed, we found that E484, F486, and F490 positions can be sensitive to Nb6 binding (Figure 5). In particular, it was experimentally determined that Nb6 binding could be severely impeded by E484K mutation [49]. We specifically examined the effect of mutations present in the S-B.1.1.7 variant (N501Y) and S-B1.351 variant (K417N, E484K, N501Y on Nb6 and VHH E binding. It appeared that K417N and N501Y mutations only moderately affected nanobody binding. Somewhat more moderate but still noticeable destabilization changes can be induced in the S trimer complexes with VHH E nanobody upon mutations of L452 and E484 sites (Figure 5). Hence, these nanobody-escaping mutations center at highly antigenic sites. The moderate stability for sites of escaping mutations is consistent with the notion that the virus tends to target positions where mutations would not appreciably perturb the RBD folding stability that is a prerequisite for proper activity of spike protein and binding with the host receptor. By targeting dynamic and structurally adaptable hotspots such as E484, F486, and F490 that are relatively tolerant to mutational changes, the virus tends to exploit conformational plasticity in these regions in eliciting specific escape patterns that would impair nanobody binding.

For the S trimer complex with VHH VE nanobody, the binding footprint revealed several clusters of binding energy hotspots (Figure 6) targeting two different epitopes. The S-RBD hotspot residues correspond to Y449, L452, F456, F486, Y489, F490, and Y508 (Figure 5A). In agreement with the experiments [49], mutations at the VHH E interface Y449H/D/N, F490S, S494P/S, G496S, and Y508H produced destabilizing ΔΔG changes exceeding 2.0 kcal/mol (Figure 6). The binding epitope for VHH V is fairly large and includes Y369, N370, S371, A372, S373, F374, F377, L378, C379, Y380, G381, V382, S383 residues. The hotspot positions in the second cryptic epitope corresponded to the conserved and stable residues Y369, S371, F374, F377, C379, Y380 (Figure 6). The escaping mutations Y369H, S371P, F374I/V, T376I, F377L, and K378Q/N at the VHH U interface resulted in considerable destabilization losses (Figure 6). Hence, flexible RBD sites F486 and F490 are consistently featured as common binding energy hotspots for these complexes, which may explain why escape mutants in these positions are known to dominate at the VHH E interface [49]. The results confirmed that nanobody combinations could alleviate the emergence and impact of escape mutants that target F456, F490, and Q493 residues.

Consistent with our earlier studies [105], mutational scanning and energetic cartography analysis suggested that VHH E/VHH V can use binding of VHH V at the cryptic binding site to form a structurally stable anchoring platform that allows for modulation of functional movements of VHH E and provides allosteric control over structural changes in the RBM epitope. Due to synergistic avidity effects, binding of the VHH E arm at the RBM epitope may then lower the entropic penalty and allow for local structural accommodations to compensate for the loss of binding interactions. This may underlie a mechanism by which multivalent nanobodies can leverage long-range couplings to synergistically inhibit distinct binding epitopes and suppress mutational escape. 

We also examined the effect of Omicron mutations in the RBD (G339D, S371L, S373P, S375F, K417N, N440K, G446S, S477N, T478K, E484A, Q493R, G496S, Q498R, N501Y, Y505H) on binding of Nb6, VHH E, and VHH E/VHH V nanobodies (Figure 7). Importantly, some of the Omicron mutations could significantly affect Nb6 binding, particularly G446S, E484A, G496S, and Y505H modifications (Figure 7A,B). The results confirmed the important role of E484 and N501 positions for protein stability and binding affinity, which is consistent with the atomic force spectroscopy studies showing the impact of mutations in these sites on binding energetics with the host receptor [78]. Recent studies also showed that Omicron mutations S477N, Q498R, and N501Y could increase ACE2 affinity anchoring the RBD to ACE2 [75]. These mutations have a moderate destabilization effect on Nb6 nanobody binding, thus potentially reducing the neutralization capacity. Moreover, it was proposed that K417N, T478K, G496S, Y505H, and the mutations at the cryptic epitope S371L, S373P, S375F can reduce affinity to ACE2 while driving immune evasion [76]. According to our data, most of these mutations, particularly G496S, Y505H, S371L, and S373P, could indeed adversely affect protein stability and binding affinity with Nb6 nanobody (Figure 7A,B). This suggests that the Omicron variant could escape the neutralization by Nb6 and this class of nanobodies with a significant overlap with the ACE2-binding site and binding epitope that includes most of the mutational sites. For VHH E binding, the large binding affinity loss resulted from E484A, Q493R, G496S, and N501Y mutations (Figure 7C,D). Importantly, these mutations are among common resistant mutations that evade many individual nanobodies [46]. Moreover, structural studies showed that Omicron mutations E484A, Q493R, and Q498R are largely responsible for immune escape from monoclonal antibodies. According to the recent study, the Omicron variant can escape the neutralization of many monoclonal antibodies, where the K417N, Q493R, and E484A Omicron mutations affect the recognition of class 1 and 2 antibodies targeting the ACE2 binding epitope [121]. Our results indicated that both Nb6 and VHH E could be sensitive to these Omicron mutations that appeared to reduce binding affinity and therefore have the potential to compromise neutralization of this class of nanobodies. These observations are consistent with the most recent study of 17 nanobodies tested against SARS-CoV-2 variants showing that efficient neutralization of the Omicron variant may be observed for synergistic nanobodies targeting multiple unique binding epitopes and exploiting conserved and cryptic epitope accessible only in the receptor-binding domain up conformation [122]. The important revelation of this analysis is appreciably smaller binding free energy changes induced by RBD-Omicron mutations in the SARS-CoV-2 S protein complex with VHH E/VHH V nanobodies (Figure 7E,F). In this case, a noticeable reduction of binding affinity was observed only for E484A, Q493R, and G496S mutations. These mutations emerged as a consistent hotspot among Omicron RBD variants that affected binding affinity with all examined nanobodies (Figure 7). It was recently shown that these mutations in the Omicron spike are compatible with the usage of diverse ACE2 orthologues for entry and could amplify the ability of the Omicron variant to infect animal species [123]. Interestingly mutations in G446, S477, T478, E484, F486 are associated with resistance to more than one monoclonal antibody, and substitutions at E484 can confer a broad resistance [124]. Moreover, mutations at the E484 position (E484A, E484G, E484D, and E484K) confer partial resistance to the convalescent plasma, showing that E484 is also one of the dominant epitopes of spike protein [123,124]. The experimental studies also showed that E484 is the “Achilles’s heel” for several important classes of antibodies and nanobodies [44,45,125]. The mutational scanning analysis supported the notion that E484A mutation can have a significant detrimental effect on nanobody binding and result in Omicron-induced escape from nanobody neutralization. 

Interestingly, our results also showed that VHH E/VHH V nanobody binding could be potentially less sensitive to Q498R, N501Y, and Y505H mutations (Figure 7E,F) as compared to binding of a single nanobody VHH E (Figure 7C,D). Accordingly, synergistic combinations of nanobodies targeting distinct binding epitopes may be more resistant to mutational escape and become less sensitive to the Omicron mutations. This is consistent with recent experiments on nanobodies and nanobody combinations, showing a remarkable ability of synergistic and especially multivalent nanobodies to combat escaping mutations through avidity-driven mechanisms between binding epitopes [53]. Moreover, the latest report of the design of a bi-paratopic nanobody, Nb1-Nb2, with high affinity and super-wide neutralization breadth against multiple variants [126]. Deep-mutational scanning experiments demonstrated that bi-paratopic Nb1-Nb2 is resistant to mutational escape against more than 60 RBD mutations and retains tight affinity and strong neutralizing activity against the Omicron virus. These illuminating experimental studies provide some support to our findings, suggesting that synergistic combinations targeting nonoverlapping epitopes on the RBD could be more effective in combating Omicron mutations than single nanobodies. It is worth noting that a broad spectrum mutational resistance of the discovered tetravalent bi-paratopic nanobody Nb1-Nb2 is significantly enhanced by exploiting unique and partially separated binding epitopes that emerged as a result of the bivalent fusion of Nb1 and Nb2 [126].

### 2.3. Perturbation Response Scanning of the SARS-CoV-2 S Complexes with Nanobodies Highlights Allosteric Role of Escaping Mutation Sites

Using the perturbation-response scanning (PRS) method [127,128,129,130,131,132,133,134], we quantified the allosteric effect of each residue in the SARS-CoV-2 complexes with a panel of studied nanobodies. The effector profiles estimate the propensities of a given residue to influence allosteric dynamic changes in other residues and are applied to identify regulatory hotspots of allosteric interactions as the local maxima along the profile. We propose that escaping variations could preferentially target structurally adaptable regulatory centers of collective movements and allosteric communications in the SARS-CoV-2 S complexes. To validate this hypothesis, we probed the allosteric effector potential of the S residues in complexes with studied nanobodies. 

The PRS effector profile for the S-RBD residues in the complex with Nb6 showed a significant overlap with the complex with ACE2 (Figure 8A,B). In the complex with Nb6, several effector peaks corresponding to structurally stable RBD regions (residues 348–352, 400–406) as well as S371, S373, V374, W436 positions from the cryptic site involved in interactions with Nb6 nanobody. The largest effector values corresponded to RBD residues Q493, G496, L452, and Y508 (Figure 8A). Notably, a number of local maxima were also aligned with the sites of escaping mutations, particularly Y449, L452, L453, F490, L492, Q493, and Y508 positions (Figure 8A). Hence, these residues can exhibit a strong allosteric potential in the complex and function as effector hotspots of allosteric signal transmission (Figure 8A,B). In contrast, sites of circulating mutations K417, E484, and N501 belong to local minima of the profile, which implies these residues are flexible sensors or transmitters of allosteric changes. This analysis also suggested that sites of escaping and circulating mutations may play a role in allosteric couplings of stable and flexible RBD regions that control signal propagation in the spike protein. While modifications of K417 and N501 residues appeared to trigger moderate changes in the binding affinity, the perturbations inflicted on these sites would have a significant effect on allosteric signaling in the complex. The results indicated that functional RBD sites might play complimentary roles in allosteric communications in the S complexes. While positions L452, Q493, G496 correspond to local maxima of the PRS profile and can assume the role of the effector regulatory points that could dispatch allosteric signals through RBD regions, other functional sites such as more flexible E484, F486, and Y501 are aligned with local minima and may act as receivers/transmitters of the allosteric signal involved in functional RBD movements. Structural mapping of allosteric effector hotspots for the S trimer complex with Nb6 nanobody revealed two clusters of residues: one cluster is in the S-RBD core region near the cryptic binding epitope, and the second cluster is near the RBM epitope (Figure 8B). These clusters form a network of functional centers that connects two binding epitopes and allow for signal transmission in the complex. It is particularly interesting given that Nb6 binds only to one of these binding epitopes. This suggests that allosteric effector centers in the RBD are allocated near the binding epitopes and are intrinsic to the S protein architecture. In this context, the pre-existing network of allosteric effector centers can be activated and modulated by nanobody binding that can exploit specific effector hotspots to allosterically propagate the binding signal to other epitopes and functional regions. We also found that the E484 site may be a critical effector hotspot for Nb6 binding. Allosteric versatility of this functional site could make it vulnerable to mutations which may alter collective dynamics and potentially be a driver of resistance to nanobodies. Indeed, mutations in the epitope centered on the E484 position (F486, F490) were shown to strongly affect neutralization for different classes of nanobodies.

The PRS profile of the S timer complex with VHH E nanobody (Figure 8C,D) featured RBD positions L452, Q493, G496, Q498, Y508 among pronounced peaks of the distribution, suggesting that these sites could function as regulatory sites of allosteric signaling in the complex. Similar to the Nb6 complex, the structural map of the effector centers highlights a cluster near the cryptic binding site of the RBD core. The overall preservation of the topology and distribution of the allosteric effector centers is evident from our analysis, supporting the notion of pre-existing regulatory control points in the S protein. Instructively, the PRS profile for the S complex with VHH VE nanobody that binds to two different binding sites revealed a partial redistribution of the allosteric centers (Figure 8E,F). In this case, the dominant, sharp peak corresponded to a cluster of residues (S371, S373, V374, F377, K378) from the cryptic site that interacts with VHH V. Smaller local peaks are associated with the RBD positions from the ACE2-binding site, primarily Q493, Q498, andY508 (Figure 8E). As a result, VHH VE binding could shift the distribution towards allosteric sites from the cryptic binding site that regulate signal propagation in the S complex, while functional residues from the RBM binding site may serve as sensors of the binding signal. The diminished dependency of allosteric signaling induced by VHH VE nanobody on the common sites of escaping mutations may be related to the effects of multimeric nanobody combinations that allow for a reduction in susceptibility to escape mutations. This suggests a plausible mechanism by which bi-paratopic nanobodies can leverage dynamic couplings to synergistically inhibit distinct binding epitopes and suppress mutational escape. To summarize, perturbation-based scanning results revealed the allosteric role of functional sites targeted by escaping mutations and the Omicron variant. Collectively, our findings suggested that SARS-CoV-2 S protein may exploit the plasticity of specific allosteric hotspots to generate escape mutants that alter response to binding without compromising activity. 

### 2.4. Network Centrality Analysis of Global Mediating Centers in the SARS-CoV-2 Complexes with Nanobodies Identifies Clusters of Allosteric Hotspots Targeted by Escaping Mutations

Network-centric models of protein structure and dynamics can allow for a more quantitative analysis of allosteric changes, identification of regulatory control centers, and mapping of allosteric communication pathways. The residue interaction networks in the SARS-CoV-2 spike trimer structures were built using a graph-based representation of protein structures [135,136] in which residue nodes are interconnected through dynamic correlations [137]. By employing network centrality calculations for the equilibrium ensembles of the SARS-CoV-2 S trimer complexes with nanobodies [138,139], we computed ensemble-averaged distributions of the short path residue centrality (Figure 9). This network metric was used to identify mediating centers of allosteric interactions in the SARS-CoV-2 complexes. In the context of the network-based centrality analysis, residues mediating a significant number of shortest pathways between all possible residue pairs in the system are identified by higher betweenness centrality. 

The network centrality profiles revealed several characteristic cluster peaks that are shared among complexes (Figure 9). However, nanobody binding can modulate this distribution and change the relative contribution of mediating centers. In the S trimer complexes with N6 and VHH E nanobodies that target the ACE2-binding sites, we observed the largest peak localized in the cluster of F490, L492, Q493, G496, Q498, and Y508 positions residues (Figure 9A,B). The second peak is aligned with Y449, L452, L453, and L455 RBD positions. In network terms, this implies that allosteric signaling in the S complexes with Nb6 and VHH E can be mediated by these sites that serve as central communication hubs. As a result, mutations in these positions and loss of interactions can affect not only the local structural environment of the mutated sites but also impact the global network organization of the system. Strikingly, a significant number of these mediating centers corresponded to residues involved in the Omicron variant. Hence, multiple Omicron RBD mutations (such as Q493R, G496S, Q498R, N501Y, Y505H) may have a measurable effect on allosteric couplings in the complexes with Nb6 and VHH E nanobodies, which would likely render some level of resistance to nanobody-induced neutralization. 

In contrast, in the S complex with bi-paratopic VHH VE nanobody, a partial redistribution of the network centrality distribution was detected, pointing to the reduced peaks in the RBD residues from the ACE2-binding site, while showing a moderate centrality for S-RBD core residues from the cryptic site (S371, F374, S375, F377, C379, Y380). The observed modulation of high centrality peaks and broadening of the distribution showed that many residues feature a moderate level of centrality. As a result, VHH VE nanobody binding can induce long-range couplings between the cryptic binding epitope and ACE2-binding site through a broader ensemble of communication paths that is less dependent on specific mediating centers and therefore may be less sensitive to mutational perturbations of functional residues. This suggests a plausible mechanism by which bi-paratopic nanobodies can leverage dynamic couplings to synergistically inhibit distinct binding epitopes and suppress mutational escape. 

## 3. Materials and Methods 

### 3.1. Structure Preparation and Analysis 

All structures were obtained from the Protein Data Bank [140,141]. During the structure preparation stage, protein residues in the crystal structures were inspected for missing residues and protons. Hydrogen atoms and missing residues were initially added and assigned according to the WHATIF program web interface [142,143]. The structures were further pre-processed through the Protein Preparation Wizard (Schrödinger, LLC, New York City, NY, USA) and included the check of bond order, assignment, and adjustment of ionization states, formation of disulfide bonds, removal of crystallographic water molecules and co-factors, capping of the termini, assignment of partial charges, and addition of possible missing atoms and side chains that were not assigned in the initial processing with the WHATIF program. The missing loops in the studied cryo-EM structures of the SARS-CoV-2 S protein were reconstructed and optimized using template-based loop prediction approaches ModLoop [144], ArchPRED server [145] and further confirmed by FALC (Fragment Assembly and Loop Closure) program [146]. The side-chain rotamers were refined and optimized by the SCWRL4 tool [147]. The conformational ensembles were also subjected to all-atom reconstruction using the PULCHRA method [148] and CG2AA tool [149] to produce atomistic models of simulation trajectories. The protein structures were then optimized using atomic-level energy minimization with composite physics and knowledge-based force fields as implemented in the 3Drefine method [150]. The atomistic structures from simulation trajectories were further elaborated by adding N-acetyl glycosamine (NAG) glycan residues and optimized. 

### 3.2. Coarse-Grained Simulations

Coarse-grained (CG) models are computationally effective approaches for simulations of large systems over long timescales. We employed a CABS-flex approach that efficiently combines a high-resolution coarse-grained model and efficient search protocol capable of accurately reproducing all-atom MD simulation trajectories and dynamic profiles of large biomolecules on a long time scale [151,152,153,154,155,156]. In this high-resolution model, the amino acid residues are represented by Cα, Cβ, the center of mass of side chains and another pseudoatom placed in the center of the Cα-Cα pseudo-bond. In this model, the amino acid residues are represented by Cα, Cβ, the center of mass of side chains and the center of the Cα-Cα pseudo-bond. The CABS-flex approach, implemented as a Python 2.7 object-oriented standalone package [154,155], was used in this study to allow for robust conformational sampling proven to accurately recapitulate all-atom MD simulation trajectories of proteins on a long time scale. Conformational sampling in the CABS-flex approach was conducted with the aid of Monte Carlo replica-exchange dynamics and involves local moves of individual amino acids in the protein structure and global moves of small fragments [151,152,153]. The default settings were used in which soft native-like restraints are imposed only on pairs of residues fulfilling the following conditions: the distance between their *C*_α_ atoms was smaller than 8 Å, and both residues belong to the same secondary structure elements. The CABS-flex default distance restraints moderately penalize the position of restrained residues if their distance differed from the distance in the original cryo-EM structure becomes more than 1 Å. In these settings, loop regions are fully unrestrained. A total of 100 independent CG-CABS simulations were performed for each of the studied systems. In each simulation, the total number of cycles was set to 10,000, and the number of cycles between trajectory frames was 100. MODELLER-based reconstruction of simulation trajectories to all-atom representation provided by the CABS-flex package was employed to produce atomistic models of the equilibrium ensembles for studied systems [121]. 

### 3.3. Molecular Dynamics Simulations 

All-atom MD simulations were performed for an N, P, T ensemble in explicit solvent using NAMD 2.13 package [157] with CHARMM36 force field [158]. Long-range non-bonded van der Waals interactions were computed using an atom-based cutoff of 12 Å with switching van der Waals potential beginning at 10 Å. Long-range electrostatic interactions were calculated using the particle mesh Ewald method [159] with a real space cut-off of 1.0 nm and a fourth-order (cubic) interpolation. SHAKE method was used to constrain all bonds associated with hydrogen atoms. Simulations were run using a leap-frog integrator with a 2 fs integration time step. Energy minimization after addition of solvent and ions was carried out using the steepest descent method for 100,000 steps. All atoms of the complex were first restrained at their crystal structure positions with a force constant of 10 Kcal mol^−1^ Å^−2^. Equilibration was done in steps by gradually increasing the system temperature in steps of 20 K starting from 10 K until 310 K, and at each step, 1ns equilibration was done, keeping a restraint of 10 Kcal mol-1 Å-2 on the protein *C*_α_ atoms. After the restraints on the protein atoms were removed, the system was equilibrated for additional 10 ns. An NPT production simulation was run on the equilibrated structures for 500 ns, keeping the temperature at 310 K and constant pressure (1 atm). In simulations, the Nose–Hoover thermostat [160] and isotropic Martyna–Tobias–Klein barostat [161] were used to maintain the temperature at 310 K and pressure at 1 atm, respectively. Principal component analysis (PCA) of MD trajectories was carried out based on the set of backbone heavy atoms using the CARMA package [162]. 

### 3.4. Mutational Scanning and Sensitivity Analysis

We conducted mutational scanning analysis of the binding epitope residues for the SARS-CoV-2 S protein complexes. Each binding epitope residue was systematically mutated using all possible substitutions, and corresponding protein stability changes were computed. The BeAtMuSiC approach [110,111,112] was employed, which is based on statistical potentials describing the pairwise inter-residue distances, backbone torsion angles, and solvent accessibilities, and considers the effect of the mutation on the strength of the interactions at the interface and on the overall stability of the complex. The binding free energy of protein-protein complex can be expressed as seen in Equation (1) by the difference in the folding free energy of the complex and folding free energies of the two protein binding partners:(1)$$\Delta G_{bind}=G^{com}-G^{A}-G^{B}$$

The change of the binding energy due to a mutation was calculated then as the following Equation (2):(2)$$\Delta \Delta G_{bind}=\Delta G_{bind}^{mut}-\Delta G_{bind}^{wt}{{}_{}}^{}$$

We leveraged rapid calculations based on statistical potentials to compute the ensemble-averaged binding free energy changes using equilibrium samples from simulation trajectories. The binding free energy changes were computed by averaging the results over 1000 equilibrium samples for each of the studied systems.

### 3.5. Perturbation Response Scanning 

The Perturbation Response Scanning (PRS) approach [127,128,129,130,131,132,133,134] follows the protocol originally proposed by Bashar and colleagues [129,130] and was described in detail in our previous studies [133]. In brief, through monitoring the response to forces on the protein residues, the PRS approach can quantify allosteric couplings and determine the protein response in functional movements. In this approach, it 3*N* × 3*N* Hessian matrix H whose elements represent second derivatives of the potential at the local minimum connect the perturbation forces to the residue displacements. The 3*N*-dimensional vector ΔR of node displacements in response to 3*N*-dimensional perturbation force follows Hooke’s law F=H∗ΔR. A perturbation force is applied to one residue at a time, and the response of the protein system is measured by the displacement vector $\Delta R\left(i\right)=H^{-1}F^{\left(i\right)}$ that is then translated into *N* × *N* PRS matrix. The second derivatives matrix H is obtained from simulation trajectories for each protein structure, with residues represented by $C_{\alpha }$ atoms and the deviation of each residue from an average structure was calculated by $\Delta R_{j}\left(t\right)=R_{j}\left(t\right)-\langle R_{j}\left(t\right)\rangle$, and corresponding covariance matrix C was then calculated by $\Delta R\Delta R^{T}$. We sequentially perturbed each residue in the SARS-CoV-2 spike structures by applying a total of 250 random forces to each residue to mimic a sphere of randomly selected directions. The displacement changes, $\Delta R^{i}$ is a 3*N*-dimensional vector describing the linear response of the protein and deformation of all the residues. Using the residue displacements upon multiple external force perturbations, we compute the magnitude of the response of residue *k* as $\langle {\left\|\Delta R_{k}^{\left(i\right)}\right\|}^{2}\rangle$ averaged over multiple perturbation forces *F*^(*i*)^, yielding the *ik*^th^ element of the *N* × *N* PRS matrix. The average effect of the perturbed effector site i on all other residues is computed by averaging over all sensors (receivers) residues j and can be expressed as $\langle {\left(\Delta R^{i}\right)}^{2}\rangle {}_{effector}$. The effector profile determines the global influence of a given residue node on the perturbations in other protein residues and can be used as proxy for detecting allosteric regulatory hotspots in the interaction networks. In turn, the j^th^ column of the PRS matrix describes the sensitivity profile of sensor residue j in response to perturbations of all residues and its average is denoted as $\langle {\left(\Delta R^{i}\right)}^{2}\rangle {}_{sensor}$. The sensor profile measures the ability of residue *j* to serve as a receiver of dynamic changes in the system.

## 4. Conclusions

In this study, we performed a comprehensive computational analysis of the SARS-CoV-2 S trimer complexes with Nb6, VHH E, and VHH E/VHH V nanobodies. We combined CG-CABS and all-atom MD simulations with binding free energy scanning, perturbation-response scanning, and network centrality analysis to examine mechanisms of nanobody-induced allosteric modulation and cooperativity in the SARS-CoV-2 S trimer complexes with nanobodies. By quantifying energetic and allosteric determinants of the SARS-CoV-2 S binding with nanobodies, we also examined nanobody-induced modulation of escaping mutations and the effect of the Omicron variant on nanobody binding. The mutational scanning analysis supported the notion that E484A mutation can have a significant detrimental effect on nanobody binding and result in Omicron-induced escape from nanobody neutralization. The results suggested that by targeting structurally adaptable hotspots such as E484, F486, and F490 that are relatively tolerant to mutational changes, the virus tends to exploit conformational plasticity in these regions in eliciting specific escape from nanobody binding. Using PRS analysis, we found that escaping mutational variants could preferentially target structurally adaptable regulatory centers of collective movements and allosteric communications in the SARS-CoV-2 S complexes. We suggested that reduced dependency of allosteric signaling induced by VHH VE nanobody on the common sites of escaping mutations may be related to the effects of multimeric nanobody combinations that allow for reduction of susceptibility to escape mutations. Our findings showed that SARS-CoV-2 S protein might exploit the plasticity of specific allosteric hotspots to generate escape mutants that alter response to binding without compromising activity. The network analysis supported these findings, showing that VHH V/VHH E nanobody binding can induce long-range couplings between the cryptic binding epitope and ACE2-binding site through a broader ensemble of communication paths that is less dependent on specific mediating centers and therefore may be less sensitive to mutational perturbations of functional residues. The results suggest that binding affinity and long-range communications of the SARS-CoV-2 complexes with nanobodies can be determined by structurally stable regulatory centers and conformationally adaptable hotspots that are allosterically coupled and collectively control resilience to mutational escape. 

## Figures and Tables

**Figure 1 ijms-23-02172-f001:**
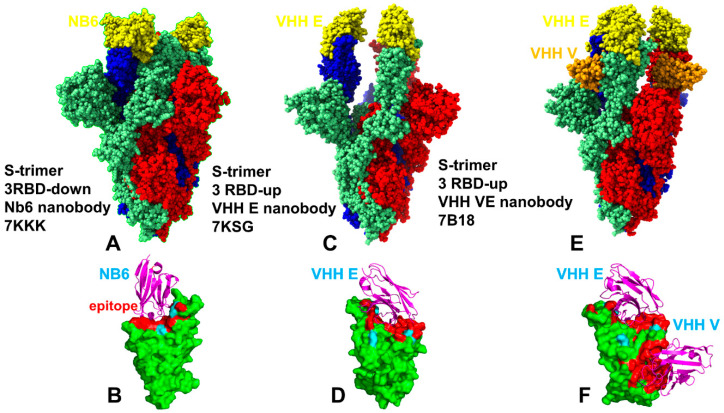
Cryo-EM structures of the SARS-CoV-2 S trimer complexes with a panel of studied nanobodies. (**A**) The structure of the SARS-CoV-2 S trimer in the complex with Nb6 nanobody, pdb id 7KKK. Nb6 nanobodies are shown in yellow spheres. (**B**) The S-RBD bound to Nb6. The S-RBD structure is shown in green surfaces. The binding epitope residues of the S-RBD bound structures are shown in red. The sites of circulating mutations K417, E484, and N501 are highlighted in cyan surfaces. Nb6 nanobody is in magenta ribbons. (**C**) The structure of the SARS-CoV-2 S trimer in the complex with VHH E nanobody, pdb id 7KSG. VHH E is in yellow spheres. (**D**) The S-RBD (in green surface) bound to VHH E (magenta ribbons). The binding epitope residues of the S-RBD bound structures are shown in red. The sites of circulating mutations K417, E484, and N501 are highlighted in cyan surfaces. (**E**) The structure of the SARS-CoV-2 S trimer in the complex with VHH E/VHH V nanobody, pdb id 7B18. VHH E is in yellow and VHH V is in orange spheres. (**F**) The S-RBD (in green surface) bound to VHH E/VHH V nanobody (magenta ribbons). The binding epitope residues of the S-RBD bound structures are shown in red. The sites of circulating mutations K417, E484, and N501 are highlighted in cyan surface. The SARS-CoV-2 S trimer structures are shown in full spheres with protomers A, B, C colored in light green, red, and blue, respectively. The rendering of SARS-CoV-2 S structures was done using the visualization program UCSF ChimeraX [108].

**Figure 2 ijms-23-02172-f002:**
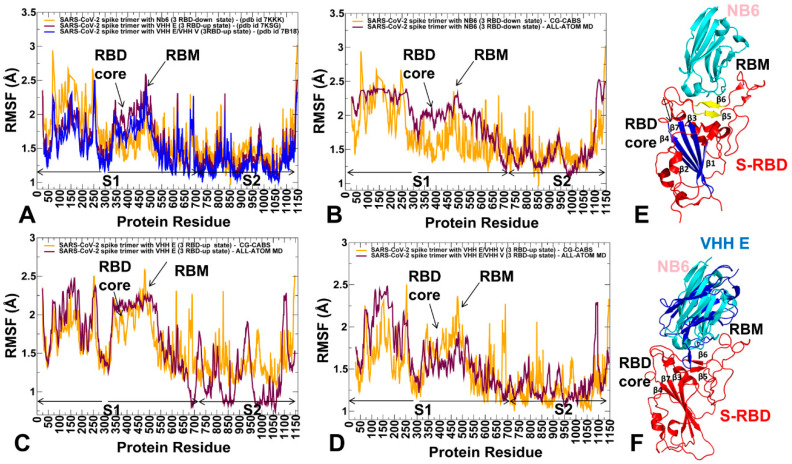
Conformational dynamics profiles of the SARS-CoV-2 S trimer complexes with nanobodies. (**A**) The root mean square fluctuations (RMSF) profiles obtained from CG-CABS simulations of the SARS-CoV-2 S trimer in the complex with Nb6 nanobody, pdb id 7KKK (in orange lines), S trimer in the complex with VHH E nanobody, pdb id 7KSG (in maroon lines), and S trimer in the complex with VHH E/VHH V nanobody, pdb id 7B18 (in blue lines). (**B**) The RMSF profiles of the SARS-CoV-2 S trimer in the complex with Nb6 obtained from CG-CABS simulations (in orange lines) and all-atom MD simulations (in maroon lines). (**C**) The R MSF profiles of the SARS-CoV-2 S trimer in the complex with VHH E obtained from CG-CABS simulations (in orange lines) and all-atom MD simulations (in maroon lines). (**D**) The RMSF profiles of the SARS-CoV-2 S trimer in the complex with VHH E/VHH V obtained from CG-CABS simulations (in orange lines) and all-atom MD simulations (in maroon lines). The position of the S-RBD core and flexible RBM regions are indicated by arrows. The S1 subunit (residues 14–685) and S2 subunit (residues 686–1163) are highlighted. The S1 domains include NTD (residues 14–306), RBD (residues 331–528), CTD1 (residues 528–591), CTD2 (residues 592–685). The S2 domains and functional regions of the simulated structures include upstream helix (UH) (residues 736–781), fusion peptide proximal region (FPPR) segment (residues 828–853), heptad repeat 1 (HR1) (residues 910–985), central helix region (CH) (residues 986–1035), connector domain (CD) (residues 1035–1068), heptad repeat 2 (HR2), (residues 1069–1163). (**E**) Structural organization of the S-RBD (shown in red ribbons). The central β strands (β1 to β4 and β7) (residues 354–358, 376–380, 394–403, 431–438, 507–516) are shown in blue. β5 and β6 strands (residues 451–454 and 492–495) are shown in yellow. The bound nanobody Nb6 is shown in cyan ribbons. (**F**) Superposition of Nb6 nanobody (in cyan ribbons) and VHH E nanobody (in blue ribbons). S-RBD is in red ribbons.

**Figure 3 ijms-23-02172-f003:**
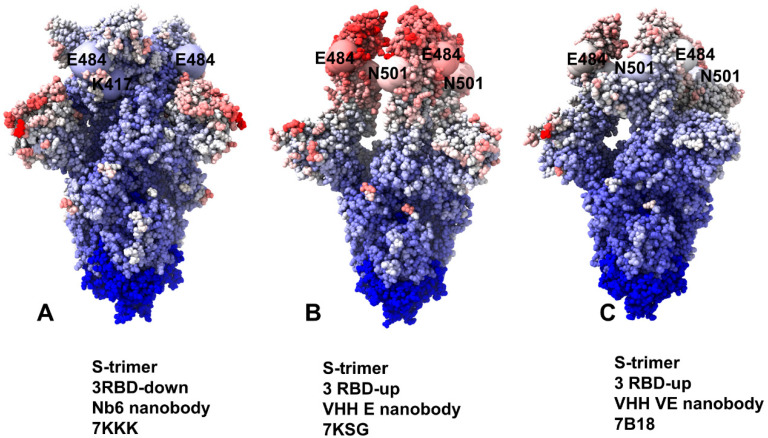
Structural maps of the conformational mobility profiles for the SARS-CoV-2 S protein variants obtained from CG-CABS simulations. The dynamics maps for the SARS-CoV-2 S trimer in the complex with Nb6 nanobody, pdb id 7KKK (**A**), S trimer in the complex with VHH E nanobody, pdb id 7KSG (**B**), and S trimer in the complex with VHH E/VHH V nanobody, pdb id 7B18 (**C**). The structures are in sphere-based representation rendered using UCSF ChimeraX [108] with the rigidity-to-flexibility sliding scale colored from blue to red. The positions of sites of circulating mutations K417, E484, and N501 are shown in large spheres and highlighted for the protomers. The structural maps are projected onto the original cryo-EM structures.

**Figure 4 ijms-23-02172-f004:**
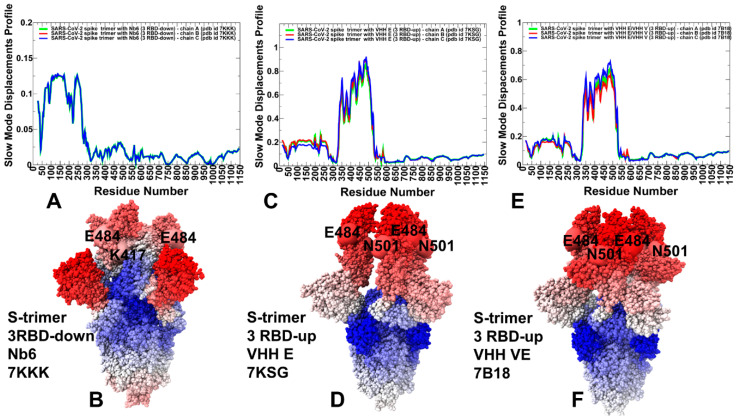
The slow mode displacement profiles of the SARS-CoV-2 S trimer structures. The low-frequency essential mobility profiles are averaged over the first three major low-frequency modes. The essential mobility profiles of the SARS-CoV-2 S trimer in the complex with Nb6 nanobody, pdb id 7KKK (**A**), S trimer in the complex with VHH E nanobody, pdb id 7KSG (**C**), and S trimer in the complex with VHH E/VHH V nanobody, pdb id 7B18 (**E**). Structural maps of the slow mode profiles for the SARS-CoV-2 S trimer in the complex with Nb6 nanobody (**B**), S trimer in the complex with VHH E nanobody, (**D**), and S trimer in the complex with VHH E/VHH V nanobody (**F**). The structures are in sphere-based representation rendered using UCSF ChimeraX [108] with the rigidity-to-flexibility sliding scale colored from blue to red. The positions of sites of circulating mutations K417, E484, and N501 are shown in large spheres and highlighted for the protomers. The structural maps are projected onto the original cryo-EM structures.

**Figure 5 ijms-23-02172-f005:**
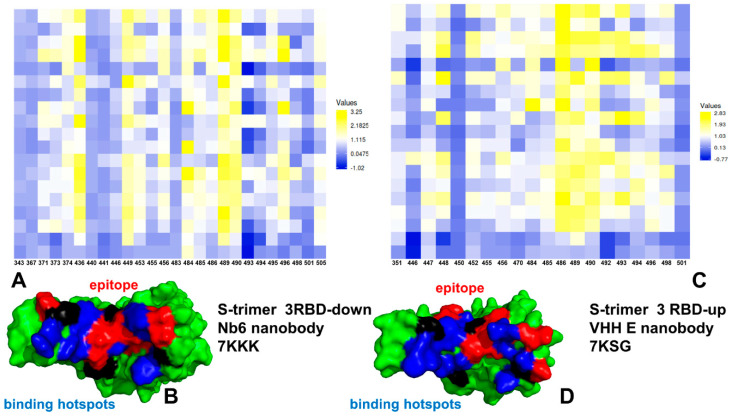
The mutational scanning heatmap for the SARS-CoV-2 S trimer complex with Nb6 nanobody, pdb id 7KKK (**A**,**B**) and VHH E nanobody, pdb id 7KSG (**C**,**D**). The binding energy hotspots correspond to residues with high mutational sensitivity. The heatmaps show the computed binding free energy changes for all single mutations on the binding epitope sites. The squares on the heatmap are colored using a 3-colored scale—from blue to yellow, with yellow indicating the largest destabilization effect. (**B**,**D**) Structural map of the binding epitopes and binding energy hotspots for Nb6 and VHH E. The S-RBD is shown in green surface. The epitope residues are shown in red and the binding energy hotspots are shown in blue surface. The computed standard errors of the mean for the binding free energy changes are based on selected samples from atomistic trajectory reconstructed from CG-CABS simulations (~1000 samples) and are within 0.0.7–0.16 kcal/mol.

**Figure 6 ijms-23-02172-f006:**
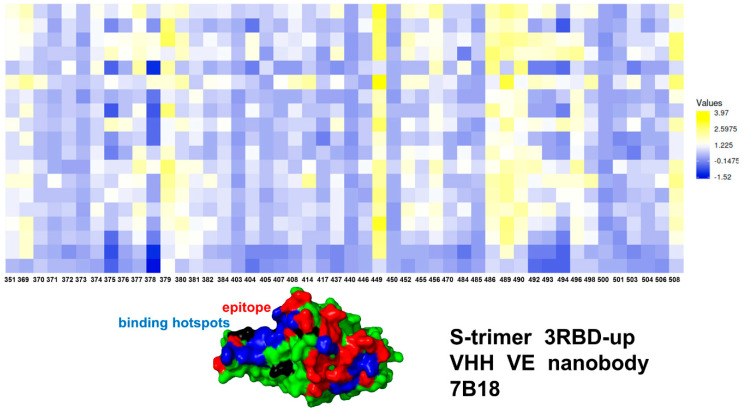
The mutational scanning heatmap for the SARS-CoV-2 S trimer complex with VHH VE nanobody, pdb id 7B18. The binding energy hotspots correspond to residues with high mutational sensitivity. The heatmaps show the computed binding free energy changes for all single mutations on the binding epitope sites. The squares on the heatmap are colored using a 3-colored scale—from blue to yellow, with yellow indicating the largest destabilization effect. Structural map of the binding epitopes and binding energy hotspots for VHH VE. The S-RBD is shown in green surface. The epitope residues are in red, and the binding energy hotspots are shown in blue surface. The computed standard errors of the mean for the binding free energy changes are based on selected samples from atomistic trajectory reconstructed from CG-CABS simulations (1000 samples) and are within 0.15–0.23 kcal/mol.

**Figure 7 ijms-23-02172-f007:**
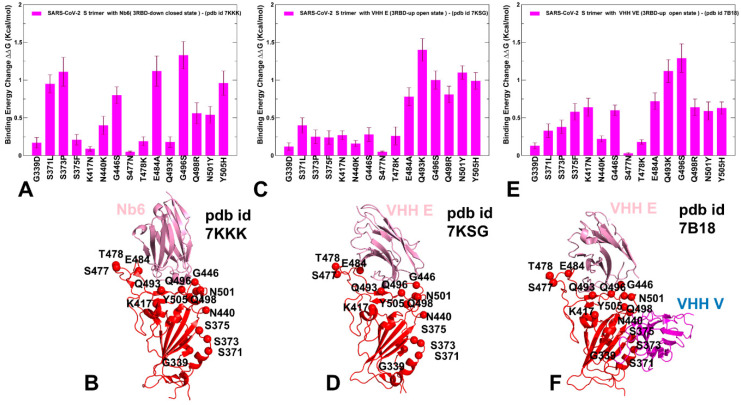
The mutational sensitivity analysis of the Omicron RBD mutations in the SARS-CoV-2 S trimer complexes with nanobodies. The binding free energy changes caused by Omicron RBD mutations on S trimer binding with Nb6 (**A**), VHH E (**C**), and VHH E/VHH V nanobody (**E**). The computed standard errors of the mean for the binding free energy changes are based on number of selected samples from atomically reconstructed CG-CABS trajectories (~1000 samples) and are generally within 0.07–0.18 kcal/mol. The error bars are shown in whisker error bar style. (**B**) Structural view of the S-RBD (in red ribbons) bound to Nb6 nanobody (in pink ribbons). (**D**) Structural view of the S-RBD (in red ribbons) bound to VHH E nanobody (in pink ribbons). (**F**) Structural view of the S-RBD (in red ribbons) bound to VHH E (in pink ribbons) and VHH V (in magenta ribbons). The positions of Omicron-RBD mutations are shown in spheres and annotated.

**Figure 8 ijms-23-02172-f008:**
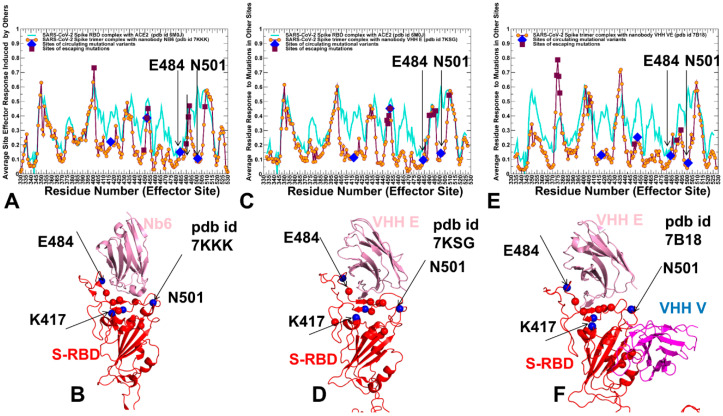
The PRS effector profiles for the SARS-CoV-2 S trimer complexes with Nb6 nanobody, pdb id 7KKK (**A**), VHH E nanobody, pdb id 7KSG (**C**), and VHH VE nanobody, pdb id 7B18 (**E**). The PRS effector profiles for the SARS-CoV-2 S complexes are shown in maroon-colored lines with orange-colored filled circles. For comparison, the PRS profiles are superimposed with the respective profiles for the S-RBD complex with ACE2 shown in cyan-colored lines (pdb id 6M0J). The sites of escaping mutations for nanobody binding are indicated by maroon-colored filled squares, and RBD sites K417, E484, and N501 targeted by global circulating variants are highlighted in blue-colored filled diamonds. The positions of sites of circulating variants E484 and N501 are indicated by arrows on panels (**A**,**C**,**E**). These sites are aligned with the local minima of the PRS profile and may act as receivers/transmitters of the allosteric signal involved in functional RBD movements. Structural maps of the allosteric effector hotspots corresponding to the local maxima of the PRS profile are shown for the SARS-CoV-2 S trimer complex with Nb6 nanobody (**B**), S trimer complex with VHH E nanobody (**D**), and S trimer complex with VHH E/VHH V nanobody (**F**). The S-RBDs are shown in red-colored ribbons rendered using UCSF ChimeraX [108]. Structural positions of allosteric effector centers are shown in red spheres. The important functional sites subjected to circulating mutations K417, E484 and N501 are shown in blue spheres. The bound nanobodies Nb6 (**B**) and VHH E (**D**) are shown in pink-colored ribbons. VHH E/VHH V nanobody is shown in pink and magenta-colored ribbons, respectively (**F**).

**Figure 9 ijms-23-02172-f009:**
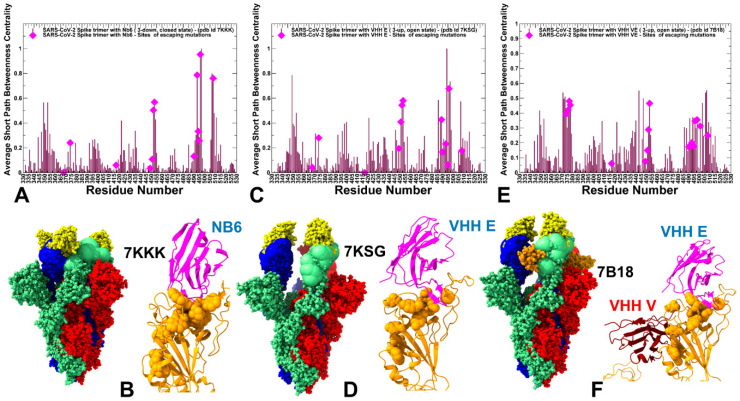
Network centrality analysis of the SARS-CoV-2 S trimer complexes with nanobodies. (**A**) The ensemble-averaged short path betweenness centrality for the SARS-CoV-2 S trimer in the complex with Nb6 nanobody, pdb id 7KKK (**A**), S trimer in the complex with VHH E nanobody, pdb id 7KSG (**C**), and S trimer in the complex with VHH E/VHH V nanobody, pdb id 7B18 (**E**). The residue-based profiles are shown for the S trimers are shown in maroon-colored filled bars. The sites of escaping mutations for nanobody binding are highlighted in blue-colored filled diamonds. (**B**) The structure of the SARS-CoV-2 S trimer in the complex with Nb6 nanobody. The S trimer is shown in full spheres with protomers A, B, C colored in light green, red, and blue, respectively. Nb6 nanobodies are shown in yellow spheres. The structural maps are projected onto the original cryo-EM structures. The rendering of SARS-CoV-2 S structures was done using the visualization program UCSF ChimeraX [108]. The sites of escaping mutations are highlighted in large spheres colored according to the respective protomer. A closeup of the S-RBD bound to Nb6. S-RBD is in orange ribbons, and Nb6 is in magenta ribbons. The sites of escaping mutations are shown in orange spheres and correspond to the highlighted positions in the centrality profile. (**D**) The structure of the SARS-CoV-2 S trimer in the complex with VHH E nanobody. VHH E is in yellow spheres, and sites of escaping mutations are shown in spheres. A closeup of the S-RBD bound to VHH E with sites of escaping mutations in orange spheres. The annotations are the same as in panel B. (**F**) The structure of the SARS-CoV-2 S trimer in the complex with VHH E/VHH V nanobody. VHH E is in yellow spheres and VHH V is in orange spheres. A closeup of the S-RBD bound to VHH E/VHH V with sites of escaping mutations in orange spheres. VHH E is in magenta ribbons, and VHH V is in red ribbons.

## Data Availability

Data is fully contained within the article. The data presented in this study are available in the article.

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
