# Peer review of "Allosteric Determinants of the SARS-CoV-2 Spike Protein Binding with Nanobodies: Examining Mechanisms of Mutational Escape and Sensitivity of the Omicron Variant"

_ijms, 2022, doi:10.3390/ijms23042172_

Round 1

Reviewer 1 Report

The study by G. Verkhivker deals with a proper combination of different computational methodologies (all-atom MD and coarse-grained MD) for the assessment of the effect of binding nanobodies onto several mutations in SARS-CoV-2 full spike (S) protein.

The study has presented a large set of simulations where the author validates the free energy in hotspot under binding of nanobodies. To my knowledge the transition from coarse-grained (CG) even at the level of CASB-flex toolkit to atomistic information could miss some important feature. The author should comment on his approach and validate the back-mapping routine and whether it fulfils energetics.

On the side of quantification of free energy (Fig 5-6), as the author has used a trajectory (~1000 data points) to calculate the change in free energy. I believe he should also provide the error in the variance.   Same is request for the Figure 7 which can be a whisker error bar style

As there is not an experimental counterpart, I wonder how the author can be so confident in the findings. One way will be to validate the approach against RBD/ACE2 for different variants of concern via SMFS (already available at Koehler, M., Ray, A., Moreira, R.A. et al. Molecular insights into receptor binding energetics and neutralisation of SARS-CoV-2 variants. Nat Commun 12, 6977 (2021). https://doi.org/10.1038/s41467-021-27325-1) and simulations. A comment and reference to this study highly valued.

Author Response

Gennady M. Verkhivker, Ph.D.

Professor Biomedical and Pharmaceutical Sciences,

Keck Center for Science and Engineering

Schmid College of Science & Technology, Chapman University

Chapman University School of Pharmacy

One University Drive, Orange CA 92866

Phone: 714-516-4586

Email: verkhivk@chapman.edu

Prof. Dr. Ian A. Nicholls  Section Editor-in-Chief

Department of Chemistry and Biomedical Sciences, Linnaeus University, Kalmar, Sweden

Prof. Dr. Vladimir N. Uversky , Section Editor-in-Chief

Molecular Medicine, University of South Florida, Tampa, USA

Special Issue "Applications of Computational Modeling in Disease, Infection and Drug Design"

Dr. Paulino Gómez-Puertas  Guest Editor

Molecular Modelling Group, Center of Molecular Biology “Severo Ochoa” (CSIC-UAM), Cantoblanco, E-28049 Madrid, Spain

Mr. Aniwat Sawangsalee

Assistant Editor

E-Mail: sawang@mdpi.com

MDPI Bangkok Office, BBD Building 12F,

626 Soi Jindatawil,Rama IV Rd. Mahaprutaram,Bang Rak,BKK 10500

Tel. (+66)97 148 8950, (+66)2 005 2299

Dear  Editors  :

Please accept the enclosed revised manuscript entitled “Allosteric  Determinants of the SARS-CoV-2 Spike Protein Binding with Nanobodies :  Examining Mechanisms of Mutational Escape and Sensitivity of the Omicron Variant ”  by  Prof. Gennady M. Verkhivker (corresponding author) for submission as a full-length research article to  International Journal of Molecular Sciences,  Special Issue "Applications of Computational Modeling in Disease, Infection and Drug Design."

 We would like to thank the referees for their detailed and  insightful critical reviews and the Editorial Manager for the extremely valuable and insightful comments that pointed us to the number of  important conceptual and organizational shortcomings in the original submission. We are also very grateful to the Referees and the Associate Editor for their general appreciation of our work, insightful suggestions and encouragement.  We have thoroughly and meticulously followed  every single of their numerous comments and recommendations and believe that an extremely  comprehensive and detailed revision of the manuscript has provided a more substantiated, well-organized and focused account of our investigation. Following the Editorial decision to recommend Major Revision and the Reviewers’ comments, we have made very significant changes in the original manuscript by (a) reorganizing and rewriting sections of the Introduction, Results and Discussion, and  Materials and Methods sections, (b)  revising and significantly improving all Figures; (c) adding a significant number of new simulations, results and analysis; (d) significantly expanding Discussion and Conclusions. These changes have been made  in order to   provide a comprehensive and satisfactory response to all the major and minor critical points and recommendations raised by the reviewers.  We have made all the requested changes and have substantially improved and focused  the manuscript by presenting a comprehensive account of the main objectives, results  and conclusions that are in full agreement with the existing experimental data.  In the revision, we have more clearly described the motivation and hypotheses behind this study, the conceptual and methodological framework and outlined the design of this study.   We have also clearly stated and substantiated the hypotheses, the major findings and conclusions of this study. We have made a clear distinction between the results and implications of this study. The central conclusions of our investigation are justified through a more rigorous and logical analysis.   We have made changes in the Introduction and completely reorganized the Results and Discussion sections to establish clear connections between the methodology, results and conclusions. 

We believe that the revised manuscript has now a better-defined focus and clear objectives, by systematically examining and verifying the main hypotheses of this study.  The central findings and conclusions of this study are now clearly formulated and extensively discussed at the end of each subsection in the Results and Discussion section. We hope that the changes in the manuscript have strengthened our general thesis and improved our work making it more rigorous and comprehensive.    We have incorporated all requested changes while maintaining a focused and logical style of the presentation. Following the recommendations of the Reviewers, we have modified all  figures, improved labeling, and edited figure captions to address recommendations of the referees and improve clarity of the presentation. In the revised version of the manuscript, we have made every effort to rectify discrepancies and typos as well as improved the logic and flow of the presentation. In the revised version of the manuscript, we have made every effort to improve the English grammar, rectify typos and inconsistencies in the text as well as streamline the logic and flow of the presentation. Finally, we have responded to all critical comment and recommendations in a cooperative and self-critical manner to improve the quality and presentation of the manuscript.  We believe that the revised manuscript fully addresses all the recommendations and the overall guidance of the Editorial Manager.

Here, we present a detailed commentary on significant changes that were made in the revision.

Major Changes:

  1. Following recommendations of Reviewers, we have significantly reorganized and rewritten Introduction and each subsection of the Results and Discussion by focusing on clear and concise formulation of the assumptions, major results and conclusions.

Following recommendations of the Reviewers, we have focused  and expanded Introduction section  by zooming on key results and objectives of this study.  Following recommendations of the Reviewers, we also made every effort to  have a more focused and substantive Introduction that allows the reader  well to the specific topic of the study.

We have expanded the introductory information  on variants of concern (VOC’s) specifically adding  the very latest structural and biophysical studies of the Omicron variant.  In particular, on page 4 of the revision we stated :

“The emergence of variants of concern (VOC’s) with the enhanced transmissibility and infectivity profile including D614G variant [57-60],  B.1.1.7 (alpha) [61-64],  B.1.351 (beta) [65,66],   B.1.1.28/P.1 (gamma) [67] and  B.1.1.427/B.1.429 (epsilon)  variants [68,69] have attracted an enormous attention in the scientific community and a  considerable variety of   the proposed mechanisms explaining functional observations from structural and biochemical perspectives.  The  detection of common mutational changes such as D614G, E484K, N501Y and K417N that are shared among  major circulating variants   B.1.1.7, B.1.351, and B.1.1.28/P.1   indicated that  these positions  can be particularly critical for modulation of the SARS-CoV-2 S protein responses. Biophysical studies   of the SARS-CoV-2 S  trimers for these variants revealed structural and  functional  effects of mutations that  can modulate dynamics and  stability of the closed and open forms, increase binding to the human receptor ACE2,  and confer immunity escape from vaccines and  different classes of monoclonal antibodies and nanobodies [70-74]. 

The recent VOC, omicron (B.1.1.529),  displaying a large number of mutations in the S-RBD regions  has further intensified  the scientific and public interest and concerns about the role and mechanisms underlying  the emergence of variants [75-79].   The latest structural and  biophysical tour-de-force investigation  convincingly demonstrated  that  Omicron-B.1.1.529 mutational  diversity can induce a widespread escape from neutralizing antibody responses [78]. According to this study,  mutations S477N, Q498R, and N501Y increase ACE2 affinity by 37-fold, serving  to anchor the RBD to ACE2,  while allowing  the  RBD region freedom to develop further mutations, including those that reduce ACE2 affinity in order to evade the neutralizing antibody response [78]. Strikingly,  K417N, T478K, G496S, Y505H, and the triple S371L, S373P, S375F  can reduce affinity to ACE2 while driving immune evasion  and  providing  a final net affinity for ACE2 similar to the original virus. Structural studies  examined several VOCs and demonstrated that Omicron variant RBD binds to human ACE2 with comparable affinity to that of the original virus [79].  The crystal and  cryo-EM structures of omicron RBD complexed with human ACE2 identified the role of key residues for receptor recognition showing that mutations E484A, Q493R, and Q493R  are responsible for immune escape  from monoclonal antibodies.”

Following recommendations of  the Reviewers,  we have also added and analyzed a series of extremely relevant biophysical studies of the SARS-CoV-2 spike proteins.

On page 4 of the revision, we stated :

“Biophysical studies provided an enormous insight into the mechanisms underlying differential binding of the S protein variants to the host  receptor ACE2 and antibodies. A series of illuminating biophysical  investigations analyzed the biophysical properties of the SARS-CoV-2 S-glycoprotein binding to ACE2  on model surfaces and on living cells using force–distance (FD) curve-based atomic force microscopy (FD-curve-based AFM)  [80,81]. By using atomic force microscopy and computer simulations, the kinetic and thermodynamic parameters  of binding between the ACE2 receptors on model surface  and S-RBD  variants (Alpha, Beta, Gamma, and Kappa) were investigated [81]. By providing  unprecedented atomistic-level details and significant insight into molecular binding mechanisms of  the SARS-CoV-2 variants, this study   observed  that the N501Y and E484Q mutations are particularly important for the greater stability, while the N501Y mutation is unlikely to significantly affect antibody neutralization [81].   By probing the interactions using AFM force spectroscopy  it was shown that the RBD mutations  in different variants  typically result in the  higher stability and affinity of the complex with ACE2 which can mediate  the increased transmissibility [81]. Moreover,  integration of biophysical experiments and molecular simulations support the idea of a stabilized interface through multiple weaker molecular interactions  that cooperatively stabilize the interface between the RBD and the ACE2 receptor.”

Finally,  for the Introduction, we have also focused our discussion and critical analysis of the relevant computational studies, particularly attempting to carefully review the available relevant studies on modeling nanobody binding with  SARS-CoV-2 spike proteins. In particular, on page 5 of the revision in the Introduction section we stated :

“Using  MD simulations and protein stability analysis we recently examined  binding of the SARS-CoV-2  RBD with single nanobodies Nb6 and Nb20,  VHH E,   a pair combination   VHH  E+U,  a bi-paratopic nanobody VHH VE, and a combination of CC12.3 antibody and VHH V/W nanobodies [108].  This study characterized  the binding energy hotspots in the SARS-CoV-2  protein and complexes with nanobodies providing a quantitative analysis  of the effects of circulating variants and escaping mutations on binding that is consistent with a broad range of  biochemical experiments.     The results suggested  that  mutational  escape  may be  controlled  through structurally adaptable binding hotspots in the receptor-accessible binding  epitope that are dynamically coupled to the stability centers in the distant binding epitope targeted by VHH U/V/W nanobodies [108].  Using computer-based design of protein–protein interactions, a number of nanobodies  were engineered in silico and selected based on the free energy landscape of protein docking verified by the recently reported cocrystal structures [109]. Another computational study examined binding mechanisms of neutralizing nanobodies targeting SARS-CoV-2 S proteins [148]. All-atom MD simulations totaling 27.6 μs in length using the recently solved structures of the RBD of SARS-CoV-2 S protein in complex with nanobodies H11-H4, H11-D4,  and  Ty1 revealed  interactions between S-RBD and the nanobodies  and estimated that the binding strength of the nanobodies to RBD is similar to that of ACE2 [110].

  1. Following recommendations of Reviewers, we have significantly updated, reorganized and rewritten each subsection of the Results and Discussion by focusing on clear and concise formulation of the assumptions, major results and conclusions.

In the revised manuscript, the rewritten Results and Discussion provide sufficient significant number of details and clearly formulates the main message of each section.  Each section in the Results and Discussion starts with clear formulation of the objectives and research design and concludes with a focused summary and conclusions.

We have made significant  improvements  and added an extremely valuable and relevant new simulation data on SARS-CoV-2 spike trimer complexes with nanobodies. The most significant and  relevant  change is that   we have   added the new results on all-atom MD simulations of the complexes that allowed to  combine and  directly compare the results of  coarse-grained and all-atom simulations.  

The work on all-atom MD simulations started at the time of the initial submission and we have mobilized all our available resources to complete these time-consuming simulations and include the results in the revision. We have also provided (a) a more detailed and substantive analysis and comparison of  coarse-grained simulations and all-atom MD simulations, (b)  presented  a comprehensive analysis  of the results and (c) discussed the results  from a quantitative  perspective and in the context of all available experimental evidence to help the reader to understand the significance and novelty of this study.

On page 6 of the revision, we stated :

“We performed multiple CG simulations of the SARS-CoV-2 S trimer protein complexes with a panel of nanobodies (Figure 1)  followed by all-atom reconstruction of  trajectories to examine how structural plasticity of the RBD regions can be modulated by binding and determine specific dynamic signatures induced by different classes of nanobodies targeting distinct binding epitopes.  All-atom MD simulations with the explicit  inclusion of the  glycosylation shield  could provide a rigorous assessment of   conformational  landscape of the SARS-CoV-2 S  proteins, such direct simulations   remain to be technically challenging due to  the size of   a complete SARS-CoV-2 S system embedded onto the membrane.  We combined CG simulations with  atomistic reconstruction  and additional optimization by adding the glycosylated microenvironment. CG-CABS trajectories were subjected to atomistic reconstruction and refinement. In addition, and for a direct comparative analysis, we also performed  all-atom MD simulations of the S trimer complexes with nanobodies. Using a comparison of CG-CABS and MD simulations, we verified the reliability of the proposed simulation model and examined how SARS-CoV-2 spike protein can exploit plasticity of the RBD regions to modulate specific dynamic responses to  nanobody binding. The conformational dynamics profiles  for CG-CABS simulations describe  the mean residue-based thermal fluctuations averaged over 100  independent CG simulations (Figure 2).  A comparative analysis of the conformational flexibility profiles for the S trimer complexes with Nb6, VHH E, and VHH E/VHH V nanobodies revealed stabilization of the interacting regions that was  particularly  strong in the  complex with the VHH E/VHH V  nanobody  pair (Figure 2A).  The  RBD core α-helical segments (residues 349-353, 405-410, and 416-423)   showed small thermal fluctuations in all complexes.  The stability of the central β strands   (residues 354-363, 389-405, and 423-436)  was especially pronounced in the  S trimer complex with Nb6 nanobody (Figures 2A).    Both CG-CABS and all-atom MD simulation models reproduced the overall stability of the conserved S-RBD core  formed by antiparallel β strands (β1 to β4 and β7) (residues 354-358, 376-380, 394-403, 431-438, 507-516) (Figures 2,3).  Atomistic MD simulations also showed  only moderate fluctuations  of β5 and β6) (residues 451-454 and 492-495) that  connect  the mobile RBM region to the central core (Figure 2).   The results showed that  Nb6  binding to the  closed conformation of the S trimer can induce a more significant stabilization of the S-RBD and RBM residues (Figure 2A). “

We are very much appreciative of these very important and insightful comments and suggestions that  pointed us to a more careful analysis and statistical assessment of the simulations. In the revision, we presented the mean RMSF distributions for spike residues that were obtained by averaging RMSF data over 100 independent simulations. The revised Figure 2 provides a complete information on the S protein dynamics and includes a detailed domain annotation of the spike protein with the residues ranges for all domains and functional regions. Following recommendations of the Reviewers, we have significantly expanded and deepened the  quantitative analysis of simulations and   performed a detailed statistical analysis  using data  from 100 independent simulations performed for each of the studied systems. We summarized the results of statistical analysis and  mean  correlations between RMSF mobility profiles for independent trajectories. The analysis allowed to substantiate  the fact that CG-CABS and all-atm MD simulations  could yield generally similar conformational dynamics profiles and reveal similar underlying dynamics trends.

On page 9 of the revision, we stated :

“To highlight similarities and differences in the mobility profiles  derived from CG-CABS and all-atom MD simulations, we performed a simple statistical analysis and computed averages and standard deviations of the RMSF values. In addition, to compare CG-CABS and all-atom MD trajectories and establish a correspondence between the dynamics profiles produced  through atomistic reconstruction of CG-CABS trajectories and all-atom  MD simulations,  we computed the average Spearman’s correlation coefficient (rs) between the respective RMSF  profiles.  Given the differences  between these simulation models,  the  correlation analysis confirmed a similar pattern of protein  flexibility yielding statistically significant correlation  rs = 0.68 for the S trimer complexes with Nb6,  rs= 0.723 for the S trimer complexes with VHH E,  and only slightly lower rs =0.624 for the complex with VHH E/VHH V nanobody. These results are similar to the outcome of the large scale validation study  that yielded   the average Spearman’s correlation coefficient  of r~ 0.7 between the RMSFs of the CG-CABS and atomistic simulations for the diverse protein set [112]. Interestingly, this study also showed  that correlations among MD trajectories obtained from different all-atom force fields could vary in a similar range (0.75- 0.82) [112].  The observed similarities of the conformational dynamics profiles suggested that CG-CABS simulations could provide a fairly accurate and affordable simulation approach for quantifying flexibility of the  SARS-CoV-2 S complexes with the panel of nanobodies. In general, our results supported the previous studies [112] indicating  that atomistic reconstruction of CG-CABS trajectories  could produce   adequate  protein flexibility profiles that are consistent with all-atom simulations  and due to a much lower cost allow for multiple independent runs and accumulation of statistically significant averages.”

Following the Reviewers’ recommendations and critique, we have considerably expanded our discussion of the mutational scanning analysis and connections between our results and the newly emerged experimental evidence on Omicron mutations and resistance patterns elicited by various nanobodies. We have introduced several very recent experimental studies on nanobody binding to SARS-CoV-2 spike variants,  including Omicron mutations that allowed  for better comparison and interpretation of the computational predictions.

On pages 15,16 of the revision, we stated :

“We also examined the effect of  Omicron mutations in the RBD (G339D, S371L, S373P, S375F, K417N, N440K, G446S, S477N, T478K, E484A, Q493R, G496S, Q498R, N501Y, Y505H) on binding of Nb6, VHH E, and VHH E/VHH V nanobodies (Figure 7). Importantly, some of the Omicron mutations could significantly affect Nb6 binding,  particularly G446S, E484A, G496S, and Y505H modifications (Figure 7A,B).  The results confirmed the important role of E484 and N501 positions for protein stability and binding affinity which is consistent with the  atomic force spectroscopy studies showing the impact of mutations in these sites on binding energetics with the host receptor [81].   Recent studies also showed that Omicron mutations S477N, Q498R, and N501Y can increase ACE2 affinity anchoring  the RBD to ACE2  [78].   These mutations have a moderate destabilization effect on Nb6 nanobody binding, thus potentially reducing the neutralization capacity. Moreover,  it was proposed that  K417N, T478K, G496S, Y505H, and the  mutations at the cryptic epitope S371L, S373P, S375F  can reduce affinity to ACE2 while driving  immune evasion [79]. According to our data,  most of these mutations, particularly G496S, Y505H, S371L, and S373P could indeed adversely affect protein stability and binding affinity with Nb6 nanobody(Figure 7A,B). This suggests that the Omicron variant  could escape the neutralization by Nb6 and this class of nanobodies  with a significant overlap with the ACE2-binding site and binding epitope that includes most the mutational sites. 

For VHH E binding, the large binding affinity loss resulted  from E484A, Q493R, G496S, and N501Y mutations (Figure 7C,D).  Importantly, these mutations are among common resistant mutations (that evade many individual nanobodies [47].  Moreover, structural studies showed that Omicron mutations E484A, Q493R, and Q498R  are largely responsible for immune escape  from monoclonal antibodies. According to the recent study, the Omicron variant  can escape the neutralization of many monoclonal antibodies, where the K417N, Q493R and E484A  Omicron mutations affect the recognition of Class 1 and 2 antibodies targeting the ACE2 binding epitope [124]. Our results indicated that both Nb6 and VHH E  could  be sensitive to these  Omicron mutations that appeared to reduce binding affinity and therefore have a potential to compromise neutralization of this class of nanobodies. These observations are consistent with the most recent study of  17 nanobodies tested against SARS-CoV-2 variants showing that efficient neutralization of the Omicron variant may be observed  for synergistic nanobodies  targeting  multiple unique binding epitopes  and exploiting  conserved and cryptic epitope accessible only in the receptor-binding domain up conformation [125]. The important revelation of this analysis are appreciably smaller binding free energy changes induced by RBD-Omicron mutations  in the SARS-CoV-2 S protein complex with VHH E/VHH V nanobodies (Figure 7E,F). In this case, a noticeable reduction of binding affinity was observed only   for E484A, Q493R  and G496S mutations.  These mutations emerged as   a consistent hotspot  among Omicron RBD variants that affected binding affinity with  all examined nanobodies (Figure 7).  It was recently shown  that these mutations  in the Omicron spike are compatible with usage of diverse ACE2 orthologues for entry and  could amplify the ability of the Omicron variant to infect animal species [127]. Interestingly mutations in G446, S477, T478, E484, F486, are associated with resistance to more than one monoclonal antibody and  substitutions at E484  can confer a  broad resistance [127].  Moreover, mutations  at E484 position (E484A, E484G, E484D, and E484K) confer a partial resistance to the convalescent plasma,  showing that  E484 is also one of the dominant epitopes of spike protein [126,127].  The experimental studies  also showed that E484 is the “Achilles’s heel”  for  several important classes of antibodies and nanobodies [45,46,128].     The mutational scanning analysis supported the notion that E484A mutation can have  a significant detrimental effect on  nanobody binding and result in Omicron-induced escape from nanobody neutralization.

Interestingly,  our results also showed that VHH E/VHH V nanobody binding cold be potentially  less sensitive to Q498R, N501Y and Y505H mutations (Figure 7E,F) as compared to binding of a single nanobody VHH E (Figure 7C,D).  Accordingly, synergistic combinations of nanobodies targeting distinct binding epitopes may be more resistant to mutational escape and  become less sensitive to the Omicron mutations.  This is consistent with recent experiments  on nanobodies and nanobody combinations,   showing a remarkable ability of synergistic and especially multivalent nanobodies to combat escaping mutations through  avidity-driven mechanisms between binding epitopes [56].  Moreover, the latest report of  the design of a bi-paratopic nanobody Nb1–Nb2, with high affinity and super-wide neutralization breadth against multiple variants [129]. Deep-mutational scanning experiments demonstrated  that bi-paratopic Nb1–Nb2  is resistant to mutational escape against more than 60 RBD  mutations and retains  tight affinity and strong neutralizing activity  against  Omicron virus.  These  illuminating experimental studies provide  some support to our  findings suggesting that synergistic combinations targeting nonoverlapping epitopes on the RBD could be more effective in combating Omicron mutations that single nanobodies. It is worth noting that a broad spectrum  mutational resistance of the discovered tetravalent bi-paratopic  nanobody Nb1-Nb2  is significantly enhanced by exploiting unique and partially separated binding epitopes emerged as a result  of the bivalent fusion of Nb1 and Nb2 [129].”

The reorganized Results and Discussion section provides a clearer  logistics  and connection of the performed simulations and analyses, offers new results and details of  the approach and simulations, as well as provides the necessary links with the experimental data whenever relevant and possible.  We also focused our Discussion on clearly formulating the original findings, the connection of our results with previous computational and experimental studies, and general implications of this work for understanding SARS-CoV-2 mutational variants.

  1. Following recommendations of the Reviewers, we have completely reorganized, redesigned and improved all Figures. In the revised manuscript, we have made numerous changes and improved design, labeling and annotation for all Figures. The results and figures are now clearly and tightly integrated and the presentation is streamlined to allow for clear understanding and justification of the results.

  1. Following major recommendations of the Reviewers, we have reorganized and improved the Materials and Methods section. In the revised version, we have added many relevant details concerning structural preparation and protocols for all-atom MD simulations. We have also condensed this section by focusing on relevant methodological details and nuances required to assess the quality of the results. We have presented a detailed account of the structures and methods used in this work.

  1. We have considerably redesigned and improved all the figures in the revised manuscript to improve the presentation and allow for a systematic analysis. We have redesigned and improved all figures, provided an extensive and detailed annotation, improved labeling of the figures and edited figure captions to address recommendations of the referees and improve clarity of the presentation.

  1. In the revised version, the hypotheses and major conclusions are more clearly formulated, and the implications of major finding are presented in more concrete and substantiated terms. In the revised version of the manuscript, we have made every effort to improve the English grammar, rectify all typos and inconsistencies in the text as well as streamline the logic and flow of the presentation. Finally, we have responded to all critical comment and recommendations in a cooperative and self-critical manner to improve the quality and presentation of the manuscript.  We believe that the revised manuscript fully addresses all the recommendations and the overall guidance of the Editorial Manager.

We believe that the revised manuscript has now a better-defined focus and clear objectives, by systematically examining and verifying the main hypotheses of this study.  The central findings and conclusions of this study are now clearly formulated and extensively discussed at the end of each subsection in the Results and Discussion section. We hope that the changes in the manuscript have strengthened our general thesis and improved our work making it more rigorous and comprehensive.    We have incorporated all requested changes while maintaining a focused and logical style of the presentation.

Now, we present the detailed responses to all Referee critique and comments:

Reviewer I Comments:

The study by G. Verkhivker deals with a proper combination of different computational methodologies (all-atom MD and coarse-grained MD) for the assessment of the effect of binding nanobodies onto several mutations in SARS-CoV-2 full spike (S) protein.

The study has presented a large set of simulations where the author validates the free energy in hotspot under binding of nanobodies. To my knowledge the transition from coarse-grained (CG) even at the level of CASB-flex toolkit to atomistic information could miss some important feature. The author should comment on his approach and validate the back-mapping routine and whether it fulfils energetics.

Author response:

First, I am extremely grateful to the Reviewer I for general appreciation of our work,  and for providing an insightful  analysis of the manuscript and a large number of extremely insightful critical assessments and recommendations pointing us to the key shortcomings, limitations and weaknesses of the original submission.  The high quality and level of details and insight that this review has provided allowed for a critical assessment and a detailed revision of the initial manuscript.  I am especially grateful for the extensive friendly critique of the manuscript that allowed us to make numerous important changes, focus on the details and provide the most comprehensive revision of the manuscript.  We have meticulously followed  each  and every single one of the Reviewers’ suggestions to properly meet significant challenges posted by this review. We have made all the recommended changes and addressed every single point (major and minor alike) in the revised manuscript to address these valid and important concerns and respective recommendations. 

We have significantly reorganized and rewritten each section of the Results and Discussion by focusing on clear and concise formulation of the assumptions, major results and conclusions. Each section in the Results starts with clear formulation of the objectives and concludes with a focused summary and conclusions. Following recommendations of the Reviewers, we have significantly expanded and deepened the  quantitative analysis of simulations  and provided a comprehensive analysis and discussed the results  from a quantitative  perspective and in the context of the existing experimental evidence.

To address this major critique, we have added the new results on all-atom MD simulations of the complexes that allowed to  combine and  directly compare the results of  coarse-grained and all-atom simulations.   The work on all-atom MD simulations started at the time of the initial submission and we have mobilized all our available resources to complete these time-consuming simulations and include the results in the revision. We have also provided (a) a more detailed and substantive analysis and comparison of  coarse-grained simulations and all-atom MD simulations, (b)  presented  a comprehensive analysis  of the results and (c) discussed the results  from a quantitative  perspective and in the context of all available experimental evidence to help the reader to understand the significance and novelty of this study.

On page 6 of the revision, we stated :

“We performed multiple CG simulations of the SARS-CoV-2 S trimer protein complexes with a panel of nanobodies (Figure 1)  followed by all-atom reconstruction of  trajectories to examine how structural plasticity of the RBD regions can be modulated by binding and determine specific dynamic signatures induced by different classes of nanobodies targeting distinct binding epitopes.  All-atom MD simulations with the explicit  inclusion of the  glycosylation shield  could provide a rigorous assessment of   conformational  landscape of the SARS-CoV-2 S  proteins, such direct simulations   remain to be technically challenging due to  the size of   a complete SARS-CoV-2 S system embedded onto the membrane.  We combined CG simulations with  atomistic reconstruction  and additional optimization by adding the glycosylated microenvironment. CG-CABS trajectories were subjected to atomistic reconstruction and refinement. In addition, and for a direct comparative analysis, we also performed  all-atom MD simulations of the S trimer complexes with nanobodies. Using a comparison of CG-CABS and MD simulations, we verified the reliability of the proposed simulation model and examined how SARS-CoV-2 spike protein can exploit plasticity of the RBD regions to modulate specific dynamic responses to  nanobody binding. The conformational dynamics profiles  for CG-CABS simulations describe  the mean residue-based thermal fluctuations averaged over 100  independent CG simulations (Figure 2).  A comparative analysis of the conformational flexibility profiles for the S trimer complexes with Nb6, VHH E, and VHH E/VHH V nanobodies revealed stabilization of the interacting regions that was  particularly  strong in the  complex with the VHH E/VHH V  nanobody  pair (Figure 2A).  The  RBD core α-helical segments (residues 349-353, 405-410, and 416-423)   showed small thermal fluctuations in all complexes.  The stability of the central β strands   (residues 354-363, 389-405, and 423-436)  was especially pronounced in the  S trimer complex with Nb6 nanobody (Figures 2A).    Both CG-CABS and all-atom MD simulation models reproduced the overall stability of the conserved S-RBD core  formed by antiparallel β strands (β1 to β4 and β7) (residues 354-358, 376-380, 394-403, 431-438, 507-516) (Figures 2,3).  Atomistic MD simulations also showed  only moderate fluctuations  of β5 and β6) (residues 451-454 and 492-495) that  connect  the mobile RBM region to the central core (Figure 2).   The results showed that  Nb6  binding to the  closed conformation of the S trimer can induce a more significant stabilization of the S-RBD and RBM residues (Figure 2A). “

Following recommendations of Reviewers, we have also performed a simple statistical analysis of the coarse-grained and all-atom MD trajectories to highlight the fact that CG-CABS simulations could produce an adequate  quantitative  picture of the conformational dynamics for the SARS-CoV-2 spike complexes with nanobodies.

We are very much appreciative of these very important and insightful comments and suggestions that  pointed us to a more careful analysis and statistical assessment of the simulations. In the revision, we presented the mean RMSF distributions for spike residues that were obtained by averaging RMSF data over 100 independent simulations. The revised Figure 2 provides a complete information on the S protein dynamics and includes a detailed domain annotation of the spike protein with the residues ranges for all domains and functional regions. Following recommendations of the Reviewers, we have significantly expanded and deepened the  quantitative analysis of simulations and   performed a detailed statistical analysis  using data  from 100 independent simulations performed for each of the studied systems. We summarized the results of statistical analysis and  mean  correlations between RMSF mobility profiles for independent trajectories. The analysis allowed to substantiate  the fact that CG-CABS and all-atm MD simulations  could yield generally similar conformational dynamics profiles and reveal similar underlying dynamics trends.

On page 9 of the revision, we stated :

“To highlight similarities and differences in the mobility profiles  derived from CG-CABS and all-atom MD simulations, we performed a simple statistical analysis and computed averages and standard deviations of the RMSF values. In addition, to compare CG-CABS and all-atom MD trajectories and establish a correspondence between the dynamics profiles produced  through atomistic reconstruction of CG-CABS trajectories and all-atom  MD simulations,  we computed the average Spearman’s correlation coefficient (rs) between the respective RMSF  profiles.  Given the differences  between these simulation models,  the  correlation analysis confirmed a similar pattern of protein  flexibility yielding statistically significant correlation  rs = 0.68 for the S trimer complexes with Nb6,  rs= 0.723 for the S trimer complexes with VHH E,  and only slightly lower rs =0.624 for the complex with VHH E/VHH V nanobody. These results are similar to the outcome of the large scale validation study  that yielded   the average Spearman’s correlation coefficient  of r~ 0.7 between the RMSFs of the CG-CABS and atomistic simulations for the diverse protein set [112]. Interestingly, this study also showed  that correlations among MD trajectories obtained from different all-atom force fields could vary in a similar range (0.75- 0.82) [112].  The observed similarities of the conformational dynamics profiles suggested that CG-CABS simulations could provide a fairly accurate and affordable simulation approach for quantifying flexibility of the  SARS-CoV-2 S complexes with the panel of nanobodies. In general, our results supported the previous studies [112] indicating  that atomistic reconstruction of CG-CABS trajectories  could produce   adequate  protein flexibility profiles that are consistent with all-atom simulations  and due to a much lower cost allow for multiple independent runs and accumulation of statistically significant averages.”

Reviewer I Comments:

On the side of quantification of free energy (Fig 5-6), as the author has used a trajectory (~1000 data points) to calculate the change in free energy. I believe he should also provide the error in the variance.   Same is request for the Figure 7 which can be a whisker error bar style.

Author response:

Following recommendations of the Reviewers, we have revised all figures and particularly improved annotations and figure captions for Figure 5,6, and 7. The computed error bars based on averaging  over 1,00  trajectory samples are  reported in figure captions for Figures 5,6.  We have completely redesigned Figure 7 by presenting  whisker-style error bars for the computed binding free energy changes induced by the S-RBD Omicron mutations. In addition, for each of the studied complexes with nanobodies, we presented a high-resolution ribbon diagram of the S-RBD bound to the respective nanobody with all RBD-Omicron mutational sites highlighted and annotated. These changes allow the reader to go through the manuscript with a clear and focused visual annotation of the spike regions and functional residues.

Reviewer I Comments:

As there is not an experimental counterpart, I wonder how the author can be so confident in the findings. One way will be to validate the approach against RBD/ACE2 for different variants of concern via SMFS (already available at Koehler, M., Ray, A., Moreira, R.A. et al. Molecular insights into receptor binding energetics and neutralization of SARS-CoV-2 variants. Nat Commun 12, 6977 (2021). https://doi.org/10.1038/s41467-021-27325-1) and simulations. A comment and reference to this study highly valued.

Author response:

We  are very  grateful for  pointing us to very important and extremely relevant references.

Following recommendations of  the Reviewers,  we have added and analyzed a series of extremely relevant biophysical studies of the SARS-CoV-2 spike proteins.

On page 4 of the revision, in the Introduction section,  we stated :

“Biophysical studies provided an enormous insight into the mechanisms underlying differential binding of the S protein variants to the host  receptor ACE2 and antibodies. A series of illuminating biophysical  investigations analyzed the biophysical properties of the SARS-CoV-2 S-glycoprotein binding to ACE2  on model surfaces and on living cells using force–distance (FD) curve-based atomic force microscopy (FD-curve-based AFM)  [80,81]. By using atomic force microscopy and computer simulations, the kinetic and thermodynamic parameters  of binding between the ACE2 receptors on model surface  and S-RBD  variants (Alpha, Beta, Gamma, and Kappa) were investigated [81]. By providing  unprecedented atomistic-level details and significant insight into molecular binding mechanisms of  the SARS-CoV-2 variants, this study   observed  that the N501Y and E484Q mutations are particularly important for the greater stability, while the N501Y mutation is unlikely to significantly affect antibody neutralization [81].   By probing the interactions using AFM force spectroscopy  it was shown that the RBD mutations  in different variants  typically result in the  higher stability and affinity of the complex with ACE2 which can mediate  the increased transmissibility [81]. Moreover,  integration of biophysical experiments and molecular simulations support the idea of a stabilized interface through multiple weaker molecular interactions  that cooperatively stabilize the interface between the RBD and the ACE2 receptor.”

On page 14 of the revision, we have discussed the relevance of this and other experimental studies in the context of  binding free energy computations of the effects of Omicron mutations on nanobody binding :

“We also examined the effect of  Omicron mutations in the RBD (G339D, S371L, S373P, S375F, K417N, N440K, G446S, S477N, T478K, E484A, Q493R, G496S, Q498R, N501Y, Y505H) on binding of Nb6, VHH E, and VHH E/VHH V nanobodies (Figure 7). Importantly, some of the Omicron mutations could significantly affect Nb6 binding,  particularly G446S, E484A, G496S, and Y505H modifications (Figure 7A,B).  The results confirmed the important role of E484 and N501 positions for protein stability and binding affinity which is consistent with the  atomic force spectroscopy studies showing the impact of mutations in these sites on binding energetics with the host receptor [81].   Recent studies also showed that Omicron mutations S477N, Q498R, and N501Y can increase ACE2 affinity anchoring  the RBD to ACE2  [78].   These mutations have a moderate destabilization effect on Nb6 nanobody binding, thus potentially reducing the neutralization capacity. Moreover,  it was proposed that  K417N, T478K, G496S, Y505H, and the  mutations at the cryptic epitope S371L, S373P, S375F  can reduce affinity to ACE2 while driving  immune evasion [79]. According to our data,  most of these mutations, particularly G496S, Y505H, S371L, and S373P could indeed adversely affect protein stability and binding affinity with Nb6 nanobody(Figure 7A,B). This suggests that the Omicron variant  could escape the neutralization by Nb6 and this class of nanobodies  with a significant overlap with the ACE2-binding site and binding epitope that includes most the mutational sites.  “

Following the Reviewers’ recommendations and critique, we have  expanded our discussion of the mutational scanning analysis and connections between our results and the newly emerged experimental evidence on Omicron mutations and resistance patterns elicited by various nanobodies. We have introduced several very recent experimental studies on nanobody binding to SARS-CoV-2 spike variants,  including Omicron mutations that allowed  for better comparison and interpretation of the computational predictions.

On pages 15,16 of the revision, we also  stated :

“For VHH E binding, the large binding affinity loss resulted  from E484A, Q493R, G496S, and N501Y mutations (Figure 7C,D).  Importantly, these mutations are among common resistant mutations (that evade many individual nanobodies [47].  Moreover, structural studies showed that Omicron mutations E484A, Q493R, and Q498R  are largely responsible for immune escape  from monoclonal antibodies. According to the recent study, the Omicron variant  can escape the neutralization of many monoclonal antibodies, where the K417N, Q493R and E484A  Omicron mutations affect the recognition of Class 1 and 2 antibodies targeting the ACE2 binding epitope [124]. Our results indicated that both Nb6 and VHH E  could  be sensitive to these  Omicron mutations that appeared to reduce binding affinity and therefore have a potential to compromise neutralization of this class of nanobodies. These observations are consistent with the most recent study of  17 nanobodies tested against SARS-CoV-2 variants showing that efficient neutralization of the Omicron variant may be observed  for synergistic nanobodies  targeting  multiple unique binding epitopes  and exploiting  conserved and cryptic epitope accessible only in the receptor-binding domain up conformation [125]. The important revelation of this analysis are appreciably smaller binding free energy changes induced by RBD-Omicron mutations  in the SARS-CoV-2 S protein complex with VHH E/VHH V nanobodies (Figure 7E,F). In this case, a noticeable reduction of binding affinity was observed only   for E484A, Q493R  and G496S mutations.  These mutations emerged as   a consistent hotspot  among Omicron RBD variants that affected binding affinity with  all examined nanobodies (Figure 7).  It was recently shown  that these mutations  in the Omicron spike are compatible with usage of diverse ACE2 orthologues for entry and  could amplify the ability of the Omicron variant to infect animal species [127]. Interestingly mutations in G446, S477, T478, E484, F486, are associated with resistance to more than one monoclonal antibody and  substitutions at E484  can confer a  broad resistance [127].  Moreover, mutations  at E484 position (E484A, E484G, E484D, and E484K) confer a partial resistance to the convalescent plasma,  showing that  E484 is also one of the dominant epitopes of spike protein [126,127].  The experimental studies  also showed that E484 is the “Achilles’s heel”  for  several important classes of antibodies and nanobodies [45,46,128].     The mutational scanning analysis supported the notion that E484A mutation can have  a significant detrimental effect on  nanobody binding and result in Omicron-induced escape from nanobody neutralization.  Interestingly,  our results also showed that VHH E/VHH V nanobody binding cold be potentially  less sensitive to Q498R, N501Y and Y505H mutations (Figure 7E,F) as compared to binding of a single nanobody VHH E (Figure 7C,D).  Accordingly, synergistic combinations of nanobodies targeting distinct binding epitopes may be more resistant to mutational escape and  become less sensitive to the Omicron mutations.  This is consistent with recent experiments  on nanobodies and nanobody combinations,   showing a remarkable ability of synergistic and especially multivalent nanobodies to combat escaping mutations through  avidity-driven mechanisms between binding epitopes [56].  Moreover, the latest report of  the design of a bi-paratopic nanobody Nb1–Nb2, with high affinity and super-wide neutralization breadth against multiple variants [129]. Deep-mutational scanning experiments demonstrated  that bi-paratopic Nb1–Nb2  is resistant to mutational escape against more than 60 RBD  mutations and retains  tight affinity and strong neutralizing activity  against  Omicron virus.  These  illuminating experimental studies provide  some support to our  findings suggesting that synergistic combinations targeting nonoverlapping epitopes on the RBD could be more effective in combating Omicron mutations that single nanobodies. It is worth noting that a broad spectrum  mutational resistance of the discovered tetravalent bi-paratopic  nanobody Nb1-Nb2  is significantly enhanced by exploiting unique and partially separated binding epitopes emerged as a result  of the bivalent fusion of Nb1 and Nb2 [129].”

In addition, we have very carefully edited the references and established a full consistency by providing full names and all authors for all references. Following guidelines, we revised the references to be consistent but in any style, provided that the consistent formatting throughout is used.   We included  author(s) name(s), journal or book title, article or chapter title (where required), year of publication, volume and issue (where appropriate) and pagination. We revised all references by providing DOI numbers for all citations.

We have thoroughly, methodically and meticulously followed  every single of  these comments and recommendations. We have addressed all the listed  omissions and typos in references and made all the requested changes as recommended. 

To conclude my response to Reviewer I comments, I am very grateful for the extensive friendly critique and detailed editing of the manuscript that allowed us to make numerous important changes, focus on the details and provide the most comprehensive revision of the manuscript.  We have meticulously followed all and every single one of the Reviewer’s suggestions to properly meet challenges posted by this review. We have made all the recommended changes and addressed every single point (major and minor alike) in the revised manuscript to address these valid and important concerns and respective recommendations. 

Summary:

In summary, we would like to express our gratitude and appreciation of the insightful and helpful reviews. We have invested a   significant amount of effort to comprehensively and meticulously address every single comment of the referee that allowed us to critically evaluate the results and improve the manuscript.  We have responded to all critical comment and recommendations in a cooperative and self-critical manner to improve the quality and presentation of the manuscript.  We hope that the revised manuscript considerably strengthened the presentation of our work as well comprehensively addressed all the concerns, comments and recommendations of the reviewers and the Editor. We believe that the replies to the referees are also sufficiently detailed showing that all referees’ suggestions and recommendations have been fully met.   

We hope therefore that the revised manuscript entitled  “Allosteric  Determinants of the SARS-CoV-2 Spike Protein Binding with Nanobodies :  Examining Mechanisms of Mutational Escape and Sensitivity of the Omicron Variant ”  by  Prof. Gennady M. Verkhivker (corresponding author) can be accepted for publication as a full-length research article in   International Journal of Molecular Sciences.

Thank you for your consideration.

Sincerely Yours,

Gennady M. Verkhivker, Ph.D.

Professor Biomedical and Pharmaceutical Sciences,

Keck Center for Science and Engineering

Schmid College of Science & Technology, Chapman University

Chapman University School of Pharmacy

One University Drive, Orange CA 92866

verkhivk@chapman.edu

http://compbiosciences.chapman.edu

Reviewer 2 Report

The manuscript is well written and organized, however there is room for improvement:

  1. In figure 1, the figure legend  for F is missing
  2. There is a lengthy description of figure 2 regarding S-RBD and cryptic epitope region. However, it is not clear where the regions lie in Figure 2A. It will be helpful for the readers if these regions are pointed  out in the RMSF profile. Also the labels on the X-axis in this figure and others need to be more clear.
  3. Line 222 the author claims that the nanobodies induce greater stability to S2 region. On what observation is this conclusion based on?
  4.  In figure 8, the author should point out residue E484 on the profiles and the ribbon diagrams shown below, since it is such a critical residue.
  5. In figure 9, it would be helpful to show the highlighted residues in on a ribbon diagram of the protein-nanobody complex, as shown in figure 8.
  6. Finally, if there has been any contribution from the author's students, they need to be acknowledged.

Author Response

Gennady M. Verkhivker, Ph.D.

Professor Biomedical and Pharmaceutical Sciences,

Keck Center for Science and Engineering

Schmid College of Science & Technology, Chapman University

Chapman University School of Pharmacy

One University Drive, Orange CA 92866

Phone: 714-516-4586

Email: verkhivk@chapman.edu

Prof. Dr. Ian A. Nicholls  Section Editor-in-Chief

Department of Chemistry and Biomedical Sciences, Linnaeus University, Kalmar, Sweden

Prof. Dr. Vladimir N. Uversky , Section Editor-in-Chief

Molecular Medicine, University of South Florida, Tampa, USA

Special Issue "Applications of Computational Modeling in Disease, Infection and Drug Design"

Dr. Paulino Gómez-Puertas  Guest Editor

Molecular Modelling Group, Center of Molecular Biology “Severo Ochoa” (CSIC-UAM), Cantoblanco, E-28049 Madrid, Spain

Mr. Aniwat Sawangsalee

Assistant Editor

E-Mail: sawang@mdpi.com

MDPI Bangkok Office, BBD Building 12F,

626 Soi Jindatawil,Rama IV Rd. Mahaprutaram,Bang Rak,BKK 10500

Tel. (+66)97 148 8950, (+66)2 005 2299

Dear  Editors  :

Please accept the enclosed revised manuscript entitled “Allosteric  Determinants of the SARS-CoV-2 Spike Protein Binding with Nanobodies :  Examining Mechanisms of Mutational Escape and Sensitivity of the Omicron Variant ”  by  Prof. Gennady M. Verkhivker (corresponding author) for submission as a full-length research article to  International Journal of Molecular Sciences,  Special Issue "Applications of Computational Modeling in Disease, Infection and Drug Design."

 We would like to thank the referees for their detailed and  insightful critical reviews and the Editorial Manager for the extremely valuable and insightful comments that pointed us to the number of  important conceptual and organizational shortcomings in the original submission. We are also very grateful to the Referees and the Associate Editor for their general appreciation of our work, insightful suggestions and encouragement.  We have thoroughly and meticulously followed  every single of their numerous comments and recommendations and believe that an extremely  comprehensive and detailed revision of the manuscript has provided a more substantiated, well-organized and focused account of our investigation. Following the Editorial decision to recommend Major Revision and the Reviewers’ comments, we have made very significant changes in the original manuscript by (a) reorganizing and rewriting sections of the Introduction, Results and Discussion, and  Materials and Methods sections, (b)  revising and significantly improving all Figures; (c) adding a significant number of new simulations, results and analysis; (d) significantly expanding Discussion and Conclusions. These changes have been made  in order to   provide a comprehensive and satisfactory response to all the major and minor critical points and recommendations raised by the reviewers.  We have made all the requested changes and have substantially improved and focused  the manuscript by presenting a comprehensive account of the main objectives, results  and conclusions that are in full agreement with the existing experimental data.  In the revision, we have more clearly described the motivation and hypotheses behind this study, the conceptual and methodological framework and outlined the design of this study.   We have also clearly stated and substantiated the hypotheses, the major findings and conclusions of this study. We have made a clear distinction between the results and implications of this study. The central conclusions of our investigation are justified through a more rigorous and logical analysis.   We have made changes in the Introduction and completely reorganized the Results and Discussion sections to establish clear connections between the methodology, results and conclusions. 

We believe that the revised manuscript has now a better-defined focus and clear objectives, by systematically examining and verifying the main hypotheses of this study.  The central findings and conclusions of this study are now clearly formulated and extensively discussed at the end of each subsection in the Results and Discussion section. We hope that the changes in the manuscript have strengthened our general thesis and improved our work making it more rigorous and comprehensive.    We have incorporated all requested changes while maintaining a focused and logical style of the presentation. Following the recommendations of the Reviewers, we have modified all  figures, improved labeling, and edited figure captions to address recommendations of the referees and improve clarity of the presentation. In the revised version of the manuscript, we have made every effort to rectify discrepancies and typos as well as improved the logic and flow of the presentation. In the revised version of the manuscript, we have made every effort to improve the English grammar, rectify typos and inconsistencies in the text as well as streamline the logic and flow of the presentation. Finally, we have responded to all critical comment and recommendations in a cooperative and self-critical manner to improve the quality and presentation of the manuscript.  We believe that the revised manuscript fully addresses all the recommendations and the overall guidance of the Editorial Manager.

Here, we present a detailed commentary on significant changes that were made in the revision.

Major Changes:

  1. Following recommendations of Reviewers, we have significantly reorganized and rewritten Introduction and each subsection of the Results and Discussion by focusing on clear and concise formulation of the assumptions, major results and conclusions.

Following recommendations of the Reviewers, we have focused  and expanded Introduction section  by zooming on key results and objectives of this study.  Following recommendations of the Reviewers, we also made every effort to  have a more focused and substantive Introduction that allows the reader  well to the specific topic of the study.

We have expanded the introductory information  on variants of concern (VOC’s) specifically adding  the very latest structural and biophysical studies of the Omicron variant.  In particular, on page 4 of the revision we stated :

“The emergence of variants of concern (VOC’s) with the enhanced transmissibility and infectivity profile including D614G variant [57-60],  B.1.1.7 (alpha) [61-64],  B.1.351 (beta) [65,66],   B.1.1.28/P.1 (gamma) [67] and  B.1.1.427/B.1.429 (epsilon)  variants [68,69] have attracted an enormous attention in the scientific community and a  considerable variety of   the proposed mechanisms explaining functional observations from structural and biochemical perspectives.  The  detection of common mutational changes such as D614G, E484K, N501Y and K417N that are shared among  major circulating variants   B.1.1.7, B.1.351, and B.1.1.28/P.1   indicated that  these positions  can be particularly critical for modulation of the SARS-CoV-2 S protein responses. Biophysical studies   of the SARS-CoV-2 S  trimers for these variants revealed structural and  functional  effects of mutations that  can modulate dynamics and  stability of the closed and open forms, increase binding to the human receptor ACE2,  and confer immunity escape from vaccines and  different classes of monoclonal antibodies and nanobodies [70-74]. 

The recent VOC, omicron (B.1.1.529),  displaying a large number of mutations in the S-RBD regions  has further intensified  the scientific and public interest and concerns about the role and mechanisms underlying  the emergence of variants [75-79].   The latest structural and  biophysical tour-de-force investigation  convincingly demonstrated  that  Omicron-B.1.1.529 mutational  diversity can induce a widespread escape from neutralizing antibody responses [78]. According to this study,  mutations S477N, Q498R, and N501Y increase ACE2 affinity by 37-fold, serving  to anchor the RBD to ACE2,  while allowing  the  RBD region freedom to develop further mutations, including those that reduce ACE2 affinity in order to evade the neutralizing antibody response [78]. Strikingly,  K417N, T478K, G496S, Y505H, and the triple S371L, S373P, S375F  can reduce affinity to ACE2 while driving immune evasion  and  providing  a final net affinity for ACE2 similar to the original virus. Structural studies  examined several VOCs and demonstrated that Omicron variant RBD binds to human ACE2 with comparable affinity to that of the original virus [79].  The crystal and  cryo-EM structures of omicron RBD complexed with human ACE2 identified the role of key residues for receptor recognition showing that mutations E484A, Q493R, and Q493R  are responsible for immune escape  from monoclonal antibodies.”

Following recommendations of  the Reviewers,  we have also added and analyzed a series of extremely relevant biophysical studies of the SARS-CoV-2 spike proteins.

On page 4 of the revision, we stated :

“Biophysical studies provided an enormous insight into the mechanisms underlying differential binding of the S protein variants to the host  receptor ACE2 and antibodies. A series of illuminating biophysical  investigations analyzed the biophysical properties of the SARS-CoV-2 S-glycoprotein binding to ACE2  on model surfaces and on living cells using force–distance (FD) curve-based atomic force microscopy (FD-curve-based AFM)  [80,81]. By using atomic force microscopy and computer simulations, the kinetic and thermodynamic parameters  of binding between the ACE2 receptors on model surface  and S-RBD  variants (Alpha, Beta, Gamma, and Kappa) were investigated [81]. By providing  unprecedented atomistic-level details and significant insight into molecular binding mechanisms of  the SARS-CoV-2 variants, this study   observed  that the N501Y and E484Q mutations are particularly important for the greater stability, while the N501Y mutation is unlikely to significantly affect antibody neutralization [81].   By probing the interactions using AFM force spectroscopy  it was shown that the RBD mutations  in different variants  typically result in the  higher stability and affinity of the complex with ACE2 which can mediate  the increased transmissibility [81]. Moreover,  integration of biophysical experiments and molecular simulations support the idea of a stabilized interface through multiple weaker molecular interactions  that cooperatively stabilize the interface between the RBD and the ACE2 receptor.”

Finally,  for the Introduction, we have also focused our discussion and critical analysis of the relevant computational studies, particularly attempting to carefully review the available relevant studies on modeling nanobody binding with  SARS-CoV-2 spike proteins. In particular, on page 5 of the revision in the Introduction section we stated :

“Using  MD simulations and protein stability analysis we recently examined  binding of the SARS-CoV-2  RBD with single nanobodies Nb6 and Nb20,  VHH E,   a pair combination   VHH  E+U,  a bi-paratopic nanobody VHH VE, and a combination of CC12.3 antibody and VHH V/W nanobodies [108].  This study characterized  the binding energy hotspots in the SARS-CoV-2  protein and complexes with nanobodies providing a quantitative analysis  of the effects of circulating variants and escaping mutations on binding that is consistent with a broad range of  biochemical experiments.     The results suggested  that  mutational  escape  may be  controlled  through structurally adaptable binding hotspots in the receptor-accessible binding  epitope that are dynamically coupled to the stability centers in the distant binding epitope targeted by VHH U/V/W nanobodies [108].  Using computer-based design of protein–protein interactions, a number of nanobodies  were engineered in silico and selected based on the free energy landscape of protein docking verified by the recently reported cocrystal structures [109]. Another computational study examined binding mechanisms of neutralizing nanobodies targeting SARS-CoV-2 S proteins [148]. All-atom MD simulations totaling 27.6 μs in length using the recently solved structures of the RBD of SARS-CoV-2 S protein in complex with nanobodies H11-H4, H11-D4,  and  Ty1 revealed  interactions between S-RBD and the nanobodies  and estimated that the binding strength of the nanobodies to RBD is similar to that of ACE2 [110].

  1. Following recommendations of Reviewers, we have significantly updated, reorganized and rewritten each subsection of the Results and Discussion by focusing on clear and concise formulation of the assumptions, major results and conclusions.

In the revised manuscript, the rewritten Results and Discussion provide sufficient significant number of details and clearly formulates the main message of each section.  Each section in the Results and Discussion starts with clear formulation of the objectives and research design and concludes with a focused summary and conclusions.

We have made significant  improvements  and added an extremely valuable and relevant new simulation data on SARS-CoV-2 spike trimer complexes with nanobodies. The most significant and  relevant  change is that   we have   added the new results on all-atom MD simulations of the complexes that allowed to  combine and  directly compare the results of  coarse-grained and all-atom simulations.  

The work on all-atom MD simulations started at the time of the initial submission and we have mobilized all our available resources to complete these time-consuming simulations and include the results in the revision. We have also provided (a) a more detailed and substantive analysis and comparison of  coarse-grained simulations and all-atom MD simulations, (b)  presented  a comprehensive analysis  of the results and (c) discussed the results  from a quantitative  perspective and in the context of all available experimental evidence to help the reader to understand the significance and novelty of this study.

On page 6 of the revision, we stated :

“We performed multiple CG simulations of the SARS-CoV-2 S trimer protein complexes with a panel of nanobodies (Figure 1)  followed by all-atom reconstruction of  trajectories to examine how structural plasticity of the RBD regions can be modulated by binding and determine specific dynamic signatures induced by different classes of nanobodies targeting distinct binding epitopes.  All-atom MD simulations with the explicit  inclusion of the  glycosylation shield  could provide a rigorous assessment of   conformational  landscape of the SARS-CoV-2 S  proteins, such direct simulations   remain to be technically challenging due to  the size of   a complete SARS-CoV-2 S system embedded onto the membrane.  We combined CG simulations with  atomistic reconstruction  and additional optimization by adding the glycosylated microenvironment. CG-CABS trajectories were subjected to atomistic reconstruction and refinement. In addition, and for a direct comparative analysis, we also performed  all-atom MD simulations of the S trimer complexes with nanobodies. Using a comparison of CG-CABS and MD simulations, we verified the reliability of the proposed simulation model and examined how SARS-CoV-2 spike protein can exploit plasticity of the RBD regions to modulate specific dynamic responses to  nanobody binding. The conformational dynamics profiles  for CG-CABS simulations describe  the mean residue-based thermal fluctuations averaged over 100  independent CG simulations (Figure 2).  A comparative analysis of the conformational flexibility profiles for the S trimer complexes with Nb6, VHH E, and VHH E/VHH V nanobodies revealed stabilization of the interacting regions that was  particularly  strong in the  complex with the VHH E/VHH V  nanobody  pair (Figure 2A).  The  RBD core α-helical segments (residues 349-353, 405-410, and 416-423)   showed small thermal fluctuations in all complexes.  The stability of the central β strands   (residues 354-363, 389-405, and 423-436)  was especially pronounced in the  S trimer complex with Nb6 nanobody (Figures 2A).    Both CG-CABS and all-atom MD simulation models reproduced the overall stability of the conserved S-RBD core  formed by antiparallel β strands (β1 to β4 and β7) (residues 354-358, 376-380, 394-403, 431-438, 507-516) (Figures 2,3).  Atomistic MD simulations also showed  only moderate fluctuations  of β5 and β6) (residues 451-454 and 492-495) that  connect  the mobile RBM region to the central core (Figure 2).   The results showed that  Nb6  binding to the  closed conformation of the S trimer can induce a more significant stabilization of the S-RBD and RBM residues (Figure 2A). “

We are very much appreciative of these very important and insightful comments and suggestions that  pointed us to a more careful analysis and statistical assessment of the simulations. In the revision, we presented the mean RMSF distributions for spike residues that were obtained by averaging RMSF data over 100 independent simulations. The revised Figure 2 provides a complete information on the S protein dynamics and includes a detailed domain annotation of the spike protein with the residues ranges for all domains and functional regions. Following recommendations of the Reviewers, we have significantly expanded and deepened the  quantitative analysis of simulations and   performed a detailed statistical analysis  using data  from 100 independent simulations performed for each of the studied systems. We summarized the results of statistical analysis and  mean  correlations between RMSF mobility profiles for independent trajectories. The analysis allowed to substantiate  the fact that CG-CABS and all-atm MD simulations  could yield generally similar conformational dynamics profiles and reveal similar underlying dynamics trends.

On page 9 of the revision, we stated :

“To highlight similarities and differences in the mobility profiles  derived from CG-CABS and all-atom MD simulations, we performed a simple statistical analysis and computed averages and standard deviations of the RMSF values. In addition, to compare CG-CABS and all-atom MD trajectories and establish a correspondence between the dynamics profiles produced  through atomistic reconstruction of CG-CABS trajectories and all-atom  MD simulations,  we computed the average Spearman’s correlation coefficient (rs) between the respective RMSF  profiles.  Given the differences  between these simulation models,  the  correlation analysis confirmed a similar pattern of protein  flexibility yielding statistically significant correlation  rs = 0.68 for the S trimer complexes with Nb6,  rs= 0.723 for the S trimer complexes with VHH E,  and only slightly lower rs =0.624 for the complex with VHH E/VHH V nanobody. These results are similar to the outcome of the large scale validation study  that yielded   the average Spearman’s correlation coefficient  of r~ 0.7 between the RMSFs of the CG-CABS and atomistic simulations for the diverse protein set [112]. Interestingly, this study also showed  that correlations among MD trajectories obtained from different all-atom force fields could vary in a similar range (0.75- 0.82) [112].  The observed similarities of the conformational dynamics profiles suggested that CG-CABS simulations could provide a fairly accurate and affordable simulation approach for quantifying flexibility of the  SARS-CoV-2 S complexes with the panel of nanobodies. In general, our results supported the previous studies [112] indicating  that atomistic reconstruction of CG-CABS trajectories  could produce   adequate  protein flexibility profiles that are consistent with all-atom simulations  and due to a much lower cost allow for multiple independent runs and accumulation of statistically significant averages.”

Following the Reviewers’ recommendations and critique, we have considerably expanded our discussion of the mutational scanning analysis and connections between our results and the newly emerged experimental evidence on Omicron mutations and resistance patterns elicited by various nanobodies. We have introduced several very recent experimental studies on nanobody binding to SARS-CoV-2 spike variants,  including Omicron mutations that allowed  for better comparison and interpretation of the computational predictions.

On pages 15,16 of the revision, we stated :

“We also examined the effect of  Omicron mutations in the RBD (G339D, S371L, S373P, S375F, K417N, N440K, G446S, S477N, T478K, E484A, Q493R, G496S, Q498R, N501Y, Y505H) on binding of Nb6, VHH E, and VHH E/VHH V nanobodies (Figure 7). Importantly, some of the Omicron mutations could significantly affect Nb6 binding,  particularly G446S, E484A, G496S, and Y505H modifications (Figure 7A,B).  The results confirmed the important role of E484 and N501 positions for protein stability and binding affinity which is consistent with the  atomic force spectroscopy studies showing the impact of mutations in these sites on binding energetics with the host receptor [81].   Recent studies also showed that Omicron mutations S477N, Q498R, and N501Y can increase ACE2 affinity anchoring  the RBD to ACE2  [78].   These mutations have a moderate destabilization effect on Nb6 nanobody binding, thus potentially reducing the neutralization capacity. Moreover,  it was proposed that  K417N, T478K, G496S, Y505H, and the  mutations at the cryptic epitope S371L, S373P, S375F  can reduce affinity to ACE2 while driving  immune evasion [79]. According to our data,  most of these mutations, particularly G496S, Y505H, S371L, and S373P could indeed adversely affect protein stability and binding affinity with Nb6 nanobody(Figure 7A,B). This suggests that the Omicron variant  could escape the neutralization by Nb6 and this class of nanobodies  with a significant overlap with the ACE2-binding site and binding epitope that includes most the mutational sites. 

For VHH E binding, the large binding affinity loss resulted  from E484A, Q493R, G496S, and N501Y mutations (Figure 7C,D).  Importantly, these mutations are among common resistant mutations (that evade many individual nanobodies [47].  Moreover, structural studies showed that Omicron mutations E484A, Q493R, and Q498R  are largely responsible for immune escape  from monoclonal antibodies. According to the recent study, the Omicron variant  can escape the neutralization of many monoclonal antibodies, where the K417N, Q493R and E484A  Omicron mutations affect the recognition of Class 1 and 2 antibodies targeting the ACE2 binding epitope [124]. Our results indicated that both Nb6 and VHH E  could  be sensitive to these  Omicron mutations that appeared to reduce binding affinity and therefore have a potential to compromise neutralization of this class of nanobodies. These observations are consistent with the most recent study of  17 nanobodies tested against SARS-CoV-2 variants showing that efficient neutralization of the Omicron variant may be observed  for synergistic nanobodies  targeting  multiple unique binding epitopes  and exploiting  conserved and cryptic epitope accessible only in the receptor-binding domain up conformation [125]. The important revelation of this analysis are appreciably smaller binding free energy changes induced by RBD-Omicron mutations  in the SARS-CoV-2 S protein complex with VHH E/VHH V nanobodies (Figure 7E,F). In this case, a noticeable reduction of binding affinity was observed only   for E484A, Q493R  and G496S mutations.  These mutations emerged as   a consistent hotspot  among Omicron RBD variants that affected binding affinity with  all examined nanobodies (Figure 7).  It was recently shown  that these mutations  in the Omicron spike are compatible with usage of diverse ACE2 orthologues for entry and  could amplify the ability of the Omicron variant to infect animal species [127]. Interestingly mutations in G446, S477, T478, E484, F486, are associated with resistance to more than one monoclonal antibody and  substitutions at E484  can confer a  broad resistance [127].  Moreover, mutations  at E484 position (E484A, E484G, E484D, and E484K) confer a partial resistance to the convalescent plasma,  showing that  E484 is also one of the dominant epitopes of spike protein [126,127].  The experimental studies  also showed that E484 is the “Achilles’s heel”  for  several important classes of antibodies and nanobodies [45,46,128].     The mutational scanning analysis supported the notion that E484A mutation can have  a significant detrimental effect on  nanobody binding and result in Omicron-induced escape from nanobody neutralization.

Interestingly,  our results also showed that VHH E/VHH V nanobody binding cold be potentially  less sensitive to Q498R, N501Y and Y505H mutations (Figure 7E,F) as compared to binding of a single nanobody VHH E (Figure 7C,D).  Accordingly, synergistic combinations of nanobodies targeting distinct binding epitopes may be more resistant to mutational escape and  become less sensitive to the Omicron mutations.  This is consistent with recent experiments  on nanobodies and nanobody combinations,   showing a remarkable ability of synergistic and especially multivalent nanobodies to combat escaping mutations through  avidity-driven mechanisms between binding epitopes [56].  Moreover, the latest report of  the design of a bi-paratopic nanobody Nb1–Nb2, with high affinity and super-wide neutralization breadth against multiple variants [129]. Deep-mutational scanning experiments demonstrated  that bi-paratopic Nb1–Nb2  is resistant to mutational escape against more than 60 RBD  mutations and retains  tight affinity and strong neutralizing activity  against  Omicron virus.  These  illuminating experimental studies provide  some support to our  findings suggesting that synergistic combinations targeting nonoverlapping epitopes on the RBD could be more effective in combating Omicron mutations that single nanobodies. It is worth noting that a broad spectrum  mutational resistance of the discovered tetravalent bi-paratopic  nanobody Nb1-Nb2  is significantly enhanced by exploiting unique and partially separated binding epitopes emerged as a result  of the bivalent fusion of Nb1 and Nb2 [129].”

The reorganized Results and Discussion section provides a clearer  logistics  and connection of the performed simulations and analyses, offers new results and details of  the approach and simulations, as well as provides the necessary links with the experimental data whenever relevant and possible.  We also focused our Discussion on clearly formulating the original findings, the connection of our results with previous computational and experimental studies, and general implications of this work for understanding SARS-CoV-2 mutational variants.

  1. Following recommendations of the Reviewers, we have completely reorganized, redesigned and improved all Figures. In the revised manuscript, we have made numerous changes and improved design, labeling and annotation for all Figures. The results and figures are now clearly and tightly integrated and the presentation is streamlined to allow for clear understanding and justification of the results.

  1. Following major recommendations of the Reviewers, we have reorganized and improved the Materials and Methods section. In the revised version, we have added many relevant details concerning structural preparation and protocols for all-atom MD simulations. We have also condensed this section by focusing on relevant methodological details and nuances required to assess the quality of the results. We have presented a detailed account of the structures and methods used in this work.

  1. We have considerably redesigned and improved all the figures in the revised manuscript to improve the presentation and allow for a systematic analysis. We have redesigned and improved all figures, provided an extensive and detailed annotation, improved labeling of the figures and edited figure captions to address recommendations of the referees and improve clarity of the presentation.

  1. In the revised version, the hypotheses and major conclusions are more clearly formulated, and the implications of major finding are presented in more concrete and substantiated terms. In the revised version of the manuscript, we have made every effort to improve the English grammar, rectify all typos and inconsistencies in the text as well as streamline the logic and flow of the presentation. Finally, we have responded to all critical comment and recommendations in a cooperative and self-critical manner to improve the quality and presentation of the manuscript.  We believe that the revised manuscript fully addresses all the recommendations and the overall guidance of the Editorial Manager.

We believe that the revised manuscript has now a better-defined focus and clear objectives, by systematically examining and verifying the main hypotheses of this study.  The central findings and conclusions of this study are now clearly formulated and extensively discussed at the end of each subsection in the Results and Discussion section. We hope that the changes in the manuscript have strengthened our general thesis and improved our work making it more rigorous and comprehensive.    We have incorporated all requested changes while maintaining a focused and logical style of the presentation.

Now, we present the detailed responses to all Referee critique and comments:

Reviewer II Comments:

The manuscript is well written and organized, however there is room for improvement:

  1. In figure 1, the figure legend  for F is missing

Author response:

I am extremely grateful to the Reviewer II for general appreciation of our work and a number of extremely insightful critical assessments and recommendations pointing us to the key shortcomings, limitations and weaknesses of the original submission.  We are grateful for the  friendly critique that allowed us to make numerous important changes, focus on the details and provide the most comprehensive revision of the manuscript.  We have meticulously followed all and every single one of the Reviewers’ suggestions to properly meet challenges posted by this review. We have made all the recommended changes and addressed every single point  in the revised manuscript to address these valid and important concerns and respective recommendations. 

The reorganized Results and Discussion section has  a better connection of the performed simulations and analyses and provides the necessary links with the experimental data whenever possible.  We also focused our Discussion on clearly formulating the original findings, the connection of our results with previous computational and experimental studies, and general implications of this work for understanding SARS-CoV-2 mutational variants.

Following recommendations of the Reviewers, we have completely  reorganized, redesigned and improved all Figures. In the revised manuscript, we have made numerous changes and improved design, labeling and annotation for all Figures.

The annotation and figure caption to the modified Figure 1 are now complete.

Reviewer II Comments:

There is a lengthy description of figure 2 regarding S-RBD and cryptic epitope region. However, it is not clear where the regions lie in Figure 2A. It will be helpful for the readers if these regions are pointed  out in the RMSF profile. Also, the labels on the X-axis in this figure and others need to be clearer.

Author response:

We have completely redesigned and reorganized Figure 2. This figure now presents the results of both CG-CABS and all-atom MD simulations.  The figure is extensively and carefully annotated and  presents a detailed visual guide of  the different referenced regions of  the spike protein.  These changes allow the reader to go through the manuscript with a clear and focused visual annotation of the spike regions and functional residues.

We have included  residue ranges and annotations for S1 and S2 subunits in the panels.  On each panel, we highlighted the rigid S-RBD core region and more flexible RBM region. In addition, the conformational dynamics profiles panels are supplemented with visual ribbon-based maps of the S-RBD regions and their binding to nanobodies. The labels on the X-axis are made larger and can be now seen more clearly.

Reviewer II Comments:

Line 222 the author claims that the nanobodies induce greater stability to S2 region. On what observation is this conclusion based on?

Author response:

We have provided (a) a more detailed and substantive analysis and comparison of  coarse-grained simulations and all-atom MD simulations, (b)  presented  a comprehensive analysis  of the results and (c) discussed the results  from a quantitative  perspective and in the context of all available experimental evidence.

Figure 2 highlights the differences in thermal fluctuations of the S1 and S2 subunits of the spike protein, showing that S2 regions are generally more stable in the complexes with nanobodies.

On page 7 of the revision, we stated :

“Interestingly, all-atom MD simulations of the SARS-CoV-2 S trimer bound to Nb6  revealed a more significant mobility of the RBD regions as compared to the conformational profile obtained in the CG-CABS simulations (Figure 2B). A greater level of flexibility  was seen in CG-CABS and atomistic MD simulations for the S-RBD regions in the S  trimer complexes with  VHH E  (Figure 2C) and VHH E/VHH V  nanobodies (Figure 2D).  Hence,  the  conformational plasticity of the RBD-up conformations  can be still maintained in the complexes with nanobodies. In comparison with all-atom MD trajectories, CG-CABS model produced the  higher average residue oscillations  which is consistent with the previous validation studies of the CABS model [112].  Consistently, both CG-CABS and all-atom MD simulations highlighted the greater stability of the  highly conserved S2 subunit (residues 686-1162) as compared to a more adaptable  S1 subunit  that includes NTD (residues 14-306), RBD (residues 331-528), CTD1 (residues 528-591), and CTD2  (residues 592-685) (Figure 2). In particular, all-atom MD simulations of the S trimer complex with VHH E nanobody showed  a  more significant difference in stabilization of the S1 and S2 domains by  displaying very small fluctuations in the  S2 regions and larger fluctuations of the S1 regions.”

Reviewer II Comments:

In figure 8, the author should point out residue E484 on the profiles and the ribbon diagrams shown below, since it is such a critical residue.

Author response:

We have redesigned and thoroughly revised all figures, including Figure 8. In the revised version, we highlighted and annotated the positions of E484 and N501 residues on the PRS profiles. In addition, we have provided detailed ribbon-based  structural maps of the S-RBD bound to nanobodies where the allosteric effector peaks are shown as red spheres, while K417, E484 and N501 are highlighted in blue spheres and annotated.

On page 16 of the revision, we discussed the PRS profiles and role of different functional sites in allosteric interactions :

“The PRS effector profile for the S-RBD residues in the  complex with Nb6 showed a significant overlap with the complex with ACE2 (Figure 8A,B) . In the complex with Nb6, several effector peaks corresponding to structurally stable RBD regions (residues 348-352, 400-406) as well as S371, S373, V374, W436 positions from the cryptic site involved in interactions with Nb6 nanobody.  The largest effector values corresponded to  RBD residues Q493, G496, L452, and Y508  (Figure 8A).  Notably, a number of local maxima were also aligned with the sites of escaping mutations, particularly Y449, L452, L453, F490, L492, Q493  and Y508 positions (Figure 8A).   Hence, these residues can exhibit a strong allosteric potential in the complex and  function as effector hotspots  of allosteric signal transmission (Figure 8A,B).   In contrast,  sites of circulating mutations  K417, E484, and N501 belong to local minima of the profile which implies these residues as flexible sensors or transmitters  of allosteric changes.  This analysis also suggested that sites of escaping and circulating mutations may play a role in allosteric couplings of stable and flexible RBD regions that control signal propagation in the spike protein.  While modifications of K417 and N501 residues appeared to trigger moderate changes in the binding affinity, the perturbations inflicted on these sites would have a significant effect on allosteric signaling in the complex. The results indicated that functional RBD sites may play complimentary roles in allosteric communications in the S complexes. While positions L452,  Q493, G496 correspond to local maxima of the PRS profile and can assume role of  the  effector regulatory points that could  dispatch allosteric signals though  RBD regions,  other functional sites such as  more flexible E484, F486 and Y501 are aligned with local minima and  may act as receivers/transmitters of  the allosteric signal  involved in functional RBD movements.” 

Reviewer II Comments:

In figure 9, it would be helpful to show the highlighted residues in on a ribbon diagram of the protein-nanobody complex, as shown in figure 8.

Author response:

Figure 9 was completely redesigned and reorganized in the revision. In addition to the network centrality profiles for studied SARS-CoV-2 spike trimer complexes with nanobodies,  the detailed structural maps for the complete trimers and S-RBD/nanobody closeups are now presented on Figure 9. The  updated figure caption provides a detailed description of the structural maps where sites of escaping mutations highlighted on the network profiles are projected accordingly to the structural maps. In figure caption we stated that the  structural maps are projected onto the original cryo-EM structures.  

Reviewer II Comments:

Finally, if there has been any contribution from the author's students, they need to be acknowledged.

Author response:

I am very proud and supportive of all my students, and they are always included in the manuscripts as co-authors when they are involved in the respective projects. In this case, it was my major individual project that I  conceived, initiated and full executed by myself from start to finish. I did make amendments in the Acknowledgement section by expressing my gratitude to students who provided minor technical assistance in preparation of this manuscript.

On page 23 of the revision, it is stated

“Acknowledgments: The author is grateful to students Keerthi Krishnan and Ryan Kassab for technical assistance in preparation of the manuscript. The author acknowledges support from  Schmid College of Science and Technology at Chapman University for providing computing resources at the Keck Center for Science and Engineering.”

To conclude my response to Reviewer II comments, I am very grateful for the extensive friendly critique and detailed editing of the manuscript that allowed us to make numerous important changes, focus on the details and provide the most comprehensive revision of the manuscript.  We have meticulously followed all and every single one of the Reviewer’s suggestions to properly meet challenges posted by this review. We have made all the recommended changes and addressed every single point (major and minor alike) in the revised manuscript to address these valid and important concerns and respective recommendations. 

Summary:

In summary, we would like to express our gratitude and appreciation of the insightful and helpful reviews. We have invested a   significant amount of effort to comprehensively and meticulously address every single comment of the referee that allowed us to critically evaluate the results and improve the manuscript.  We have responded to all critical comment and recommendations in a cooperative and self-critical manner to improve the quality and presentation of the manuscript.  We hope that the revised manuscript considerably strengthened the presentation of our work as well comprehensively addressed all the concerns, comments and recommendations of the reviewers and the Editor. We believe that the replies to the referees are also sufficiently detailed showing that all referees’ suggestions and recommendations have been fully met.   

We hope therefore that the revised manuscript entitled  “Allosteric  Determinants of the SARS-CoV-2 Spike Protein Binding with Nanobodies :  Examining Mechanisms of Mutational Escape and Sensitivity of the Omicron Variant ”  by  Prof. Gennady M. Verkhivker (corresponding author) can be accepted for publication as a full-length research article in   International Journal of Molecular Sciences.

Thank you for your consideration.

Sincerely Yours,

Gennady M. Verkhivker, Ph.D.

Professor Biomedical and Pharmaceutical Sciences,

Keck Center for Science and Engineering

Schmid College of Science & Technology, Chapman University

Chapman University School of Pharmacy

One University Drive, Orange CA 92866

verkhivk@chapman.edu

http://compbiosciences.chapman.edu

Round 2

Reviewer 1 Report

I appreciate the massive effort carried out by the authors. My suggestions have been extensively bee addressed in the revised version. I endorse the article for publication in IJMS and wish all successes to the author.